# Immunotranscriptomic profiling the acute and clearance phases of a human challenge dengue virus serotype 2 infection model

John P. Hanley [1,2], Huy A. Tu[1,3,4], Julie A. Dragon [1,5], Dorothy M. Dickson [1,2,4], Roxana del Rio-Guerra[6], Scott W. Tighe [5], Korin M. Eckstrom [1,5], Nicholas Selig[4], Samuel V. Scarpino[7], Stephen S. Whitehead [8], Anna P. Durbin [9], Kristen K. Pierce[1,2,4], Beth D. Kirkpatrick[1,2,4], Donna M. Rizzo[10], Seth Frietze [3,11,12] & Sean A. Diehl [1,2,3,4 ✉]

About 20–25% of dengue virus (DENV) infections become symptomatic ranging from self-limiting fever to shock. Immune gene expression changes during progression to severe dengue have been documented in hospitalized patients; however, baseline or kinetic information is difficult to standardize in natural infection. Here we profile the host immunotranscriptome response in humans before, during, and after infection with a partially attenuated rDEN2Δ30 challenge virus (ClinicalTrials.gov NCT02021968). Inflammatory genes including type I interferon and viral restriction pathways are induced during DENV2 viremia and return to baseline after viral clearance, while others including myeloid, migratory, humoral, and growth factor immune regulation factors pathways are found at non-baseline levels post-viremia. Furthermore, pre-infection baseline gene expression is useful to predict rDEN2Δ30-induced immune responses and the development of rash. Our results suggest a distinct immunological profile for mild rDEN2Δ30 infection and offer new potential biomarkers for characterizing primary DENV infection.

[1] Department of Microbiology and Molecular Genetics, Larner College of Medicine, University of Vermont, Burlington, VT, USA. [2] Translational Global Infectious Disease Research Center, Larner College of Medicine, University of Vermont, Burlington, VT, USA. [3] Cellular and Molecular Biomedical Sciences Program, University of Vermont, Burlington, VT, USA. [4] Vaccine Testing Center, Larner College of Medicine, University of Vermont, Burlington, VT, USA. [5] Vermont Integrated Genomics Resource, University of Vermont, Burlington, VT, USA. [6] Flow Cytometry and Cell Sorting Facility, Department of Surgery, Larner College of Medicine, University of Vermont, Burlington, VT, USA. [7] Network Science Institute, Northeastern University, Boston, MA, USA. [8] Laboratory of Viral Diseases, National Institute of Allergy and Infectious Diseases, Bethesda, MD, USA. [9] Center for Immunization Research, Department of International Health, Bloomberg School of Public Health, Johns Hopkins University, Baltimore, MD, USA. [10] Department of Civil and Environmental Engineering, College of Engineering and Mathematical Sciences, University of Vermont, Burlington, VT, USA. [11] Department of Biomedical and Health Sciences, College of Nursing and Health Sciences, University of Vermont, Burlington, VT, USA. [12] University of Vermont Cancer Center, Burlington, VT, USA. ✉email: sean.diehl@med.uvm.edu

Dengue is the number one arthropod-borne viral infection in terms of worldwide infections, with nearly 400 million occurring annually[1]. Dengue disease occurs in 96 million people per year and symptoms range from a self-limiting fever with rash, arthralgia, myalgia, fatigue, and retro-orbital pain to dengue hemorrhagic fever or dengue shock syndrome[2,3]. *Aedes spp.* mosquitos carry any of the four dengue virus serotypes (DENV1-4) which circulate in unpredictable patterns and each can cause disease[4,5]. Primary infection with a single serotype can cause disease, but the preponderance of disease is associated with secondary heterologous infection[6]. One hypothesis for this involves a reduced ratio of protective neutralizing antibodies versus non-neutralizing cross-reactive antibodies with the latter promoting antibody-dependent enhancement of viral replication as discussed elsewhere[7,8]. It is also possible that additional immune mechanisms elicited by primary DENV infection are associated with secondary disease risk[9].

Several studies have been performed in dengue natural infection cohorts using whole blood[10–18] and single cell[19] transcriptomics towards identification of immune signatures of dengue disease. Nearly all these study designs utilized symptomatic infection cases (often hospitalized patients) and correlated gene expression with clinical status through severe disease to convalescence. A recent meta-analysis of these studies has identified a discrete natural killer (NK) cell-associated 20-gene set signature of severe disease[20]. However, much less is known regarding the global changes that occur in the blood immune compartment after primary dengue infection. Similarly, there are no immunotranscriptome data sets for known mild primary dengue infections including baseline gene expression. These are important knowledge gaps to address for several reasons. First, most dengue disease is due to secondary infections[4] which, by definition, arise only after an antecedent primary infection. Second, serotype-specific differences in primary dengue disease and immunity have been described[21]. Third, it is estimated that over 75% of dengue infections do not develop into clinically apparent dengue disease[1] many of which could be primary infections, but blood gene expression changes have not been studied in such cases. Lastly, existing study designs in natural infection do not allow for a comparison of preinfection baseline with post-infection convalescence status.

As part of the ongoing effort to develop a broadly effective dengue vaccine, live DENV strains have been developed to assess the protective efficacy of vaccine candidates against experimental challenge[22–25]. rDEN2Δ30 is a recombinant serotype 2 virus based on the American genotype 1974 Tonga DENV2 virus[26,27] that has been partially attenuated by deletion of 30 nucleotides in the 3′ untranslated region of the RNA genome (Δ30)[28]. rDEN2Δ30 infection induces modest viremia in all flavivirus-naive subjects and a mild, transient non-pruritic rash in 80% of recipients[29]. We previously showed that the T-cell response to rDEN2Δ30 is comparable to that observed in natural infection[30]. Taken together with the good clinical tolerability, these data suggest that rDEN2Δ30 infection could be a suitable model for a mild and largely asymptomatic primary DENV2 infection.

To assess whether rDEN2Δ30 infection in humans induced transient or prolonged systemic changes in blood transcriptome that may correlate with clinical infection or development of immune memory we performed a differential gene expression analysis on whole blood transcriptomes captured at baseline, during infection, and after viral clearance in primary infection of dengue-naive subjects with rDEN2Δ30. Our temporal analyses revealed distinct sets of genes which directly tracked with viremia as well as those distinguishing pre- and postinfection immune states. Furthermore, we identified suites of baseline genes which correlated to subsequent clinical lab findings or protection from rash stemming from rDEN2Δ30 infection.

## Results

### Virological and clinical laboratory course of rDEN2Δ30 infection.

Infection of flavivirus-naive subjects with a rDEN2Δ30 induced viremia within 2 weeks. The only clinical symptom was a transient, mild, non-pruritic rash in 80% of subjects[29]. To investigate gene expression during experimental primary DENV infection, we performed RNA-seq on whole blood from rDEN2Δ30-infected subjects at 0, 8, and 28 days post infection. From our parent cohort of 20 subjects infected with rDEN2Δ30, including both men and women of White or Black race across two different study sites[29], we selected a representative subset of eleven based on viremia onset, duration, and peak titer metrics that matched the variability (quantified as Shannon Entropy) in these metrics of the parent cohort. This was done to understand how gene expression may change as a function of viral replication characteristics as opposed to studying the role of host background (race or sex), neither of which has to date been conclusively found to be a determinant of dengue disease. Through the transcriptomics subset was chosen based solely on variability in viremia traits, we also confirmed that there was no difference compared to the parent cohort in terms of demographics (sex, race, and study site, Supplementary Table 1) or viral load, onset, or duration viremia characteristics (Fig. 1a and Supplementary Fig. 1a). Subject-specific viremia courses of the transcriptomic subset are shown in Supplementary Fig. 1b.

All subjects in the parent study and transcriptomic subset seroconverted to DENV2 within 6 months after infection (Fig. 1b and Supplementary Fig. 2a, b), albeit to different titers in terms of strain and by individual subject (Supplementary Fig. 2c). There was no difference in serum neutralizing antibody titers against the infecting genotype 6 months post infection between the parent and transcriptomics cohort (Supplementary Fig. 2).

rDEN2Δ30 was previously found to be well-tolerated[29] though the kinetics and magnitude of the clinical laboratory findings has not been detailed. We therefore assessed immunological, hematological, clotting, and metabolic, parameters before and after rDEN2Δ30 infection. There were transient alterations in white blood cell (WBC) counts, with four subjects' WBC going slightly below the lower limit of the normal range, but in all subjects WBC returned to preinfection baselines (Fig. 2a). See Source Data for subject-specific responses. Within the transcriptomics cohort there appeared to be two groups of subjects with differing preinfection WBC levels (those ~8000 and ~4000 cells/$cm^2$), however, there was no difference in peak viremia between the groups ($P = 0.41$). RBC counts were not affected by rDEN2Δ30 infection. Basophils and eosinophils stayed within limits and were not affected by infection. Lymphocytes initially decreased after infection and rebounded, and five subjects exhibited mild lymphopenia for only 2–4 days. Monocytes trended higher after infection but stayed within normal limits. Neutrophils most closely tracked the WBC values that initially dropped after infection and five subjects exhibited transient neutropenia (ANC < 1500/$cm^2$). One subject had benign neutropenia that was unaffected by rDEN2Δ30 infection. Thus, within the WBC compartment, rDEN2Δ30 infection led to a brief decrease in neutrophils and lymphocytes, while monocytes increased before returning to baseline. Hematologic parameters stayed within normal limits, while two subjects had low mean corpuscular hemoglobin concentration though not due to infection. Hematocrit (HCT) and hemoglobin (Hb) were not affected by rDEN2Δ30 infection though two subjects had low HCT and Hb and one just low Hb (Fig. 2b). For the clotting parameters, there were overall mild drops in platelets after infection, though not to critical levels (still over 100,000/$cm^2$) (Fig. 2c). Prothrombin time and partial thromboplastin time (PTT) were not affected by infection and stayed within normal limits. Metabolic labs were unaffected by rDEN2Δ30 infection (Fig. 2d). Overall, these results demonstrated that rDEN2Δ30-induced reproducible but modest viremia and a

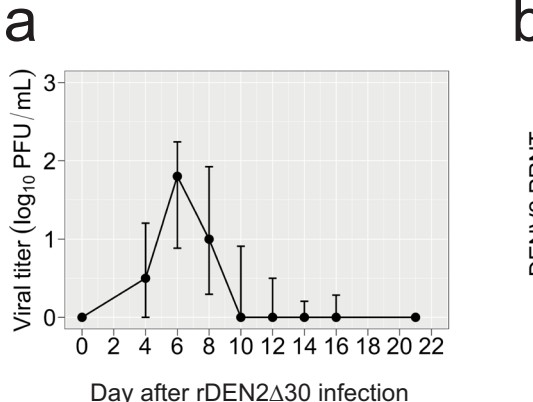

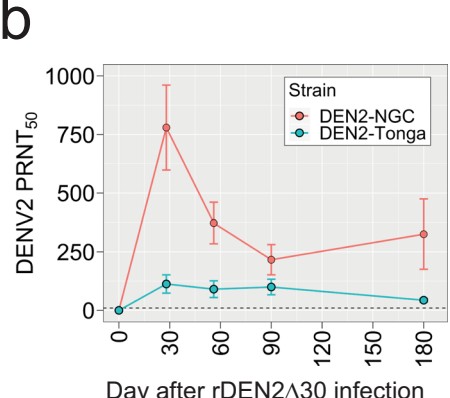

**Fig. 1 Viremia and antibody levels in response to experimental infection of humans with rDEN2Δ30 (Tonga 74 strain). a** DENV2 viremia in a cohort of flavivirus-naïve subjects, $n = 11$. Viremia was measured by culture of serum on Vero81 cells and mean viral titer plaque-forming units per mL of serum (PFU/mL) ± standard deviation (approx. 67th percentile) above zero are shown. **b** Serum neutralizing antibodies (PRNT$_{50}$, mean ± standard deviation) to DENV2-NGC and DENV2-Tonga after infection with rDEN2Δ30.

mild rash as the only clinically significant finding in DENV-naive subjects, thus approximating a clinically mild primary infection.

**Immune gene and cell dynamics during rDEN2Δ30 infection.** Dengue elicits viremic and post-viremic phases, with the latter paradoxically being associated with progression to vascular leak and severe disease[31]. To ask whether distinct immune profiles manifest during and after rDEN2Δ30 viremia we performed RNA Seq analysis of globin-depleted whole blood in these 11 subjects at days 0, 8, and 28 after rDEN2Δ30 infection, corresponding to baseline, acute viremia, and post-viremia timepoints. RNA quantity and quality was similar across timepoints (Supplementary Fig. 3). Principal component analysis of variable gene expression reveals an overall similar pattern of response for samples of corresponding timepoints with minimal overlap between baseline (day 0) and peak viremia (day 8). The day 28 data (post viremia) partially overlapped with the baseline (day 0) and acute (day 8) timepoints but exhibited a degree of separation from the day 0 data (Fig. 3a). Unsupervised hierarchical clustering of the top 2400 variable genes distinguished the different timepoints on a subject-by-subject basis (Fig. 3b). We next determined differential expression analysis in pairwise comparisons by timepoint. There were 4212 differentially expressed genes (DEGs) at day 8 post-infection compared to baseline; 1915 of which were regulated more than 1.5-fold. There were 359 DEGs between viremia and post-viremia (day 28 vs. 8); all of which were regulated more than 1.5-fold. Lastly, there were 3169 DEGs between post-viremia and baseline (day 28 vs. 0), 1064 of which were regulated more than 1.5-fold. Pathways enriched in the type I and type II interferon and antiviral responses were upregulated in viremia compared to baseline samples, whereas pathways controlling translational initiation were downregulated (Fig. 3c, d). Between viremia and post-viremia, interferon and antiviral defense responses are downregulated. We found specific downregulation of pathways involved in innate immune regulation after viral clearance. We further found that the expression of genes involved in NF-κB and IL-17 signaling pathways were significantly different post-viremia compared to baseline. In addition, apoptosis, toll-like receptor signaling, response to viruses, ribosomes, and defense responses were also differentially regulated post-viremia compared to baseline. Taken together these data showed that primary rDEN2Δ30 infection elicited a broad range of gene expression changes, some of which tracked closely with viral replication and others that may be involved in the establishment of immune memory. These results indicated

sets of genes which closely tracked with viremia and other sets that exhibited regulation that was not strictly coupled to the viral replication period.

To further explore immune system changes during and after rDEN2Δ30 infection we applied a deconvolution approach that was developed for tissues[32] but also been used to investigate blood immune changes during malaria infection[33]. Myeloid cells including monocytes and activated dendritic cells increased during acute infection and returned to baseline (Fig. 4a), which agrees with clinical lab data showing monocyte changes during infection (Fig. 2a). Activated NK cells trended higher during acute infection before returning to baseline. In the adaptive immune compartment, regulatory T cells (Tregs) were significantly ($P = 0.023$) affected by rDEN2Δ30 infection, decreasing during acute stage and manifesting a higher median frequency at convalescence compared to baseline. Naive B cells were weakly reduced during acute infection only. CD4$^{+}$, and CD8$^{+}$ T cells, plasma cells did not change during rDEN2Δ30 infection (Fig. 4a). In sum our bioinformatic analysis of immune-cell composition indicated that activated myeloid and innate immune responses tracked directly with viremia and that an elevated Treg signature persisted after clearance of virus.

To corroborate our deconvolution analysis, we performed multi-parametric flow cytometry on peripheral blood mononuclear cells (PBMC) from nine of the subjects for which samples corresponding to baseline, day 8, and days 28 were available. See Supplementary Table 2, Supplementary Fig. 4, and Supplementary Fig. 5 for conjugated antibodies used, gating and development of the staining panel, respectively. Activated HLA-DR$^{hi}$ dendritic cells (DC) HLA-DR$^{hi}$ (Fig. 4b) and CD14$^{+}$CD16$^{low}$ classical interleukin (IL)-10-producing monocytes[34] were affected by rDEN2Δ30 infection (Fig. 4c). No changes in NKT cells or in defined NK subpopulations were found (Fig. 4d), though NK changes were weak in the deconvolution analysis. The T-cell compartment overall was not affected by rDEN2Δ30 infection (Fig. 4e), though in the CD8$^{+}$ T-cell compartment, cells of the effector memory re-expressing CD45RA (T$_{EMRA}$) and central memory (T$_{CM}$) phenotypes were found to be significantly regulated by infection, with weaker effects on cytotoxic, effector memory (T$_{EM}$) and naive CD8$^{+}$ T cells (Fig. 4f). In the CD4$^{+}$ T-cell compartment, recently activated CD279 (PD-1)$^{+}$ cells as well as those with T$_{EM}$ and T$_{CM}$ phenotypes were affected by rDEN2Δ30 infection (Fig. 4g). Both CD4 and CD8 T-cell activation have been shown in dengue natural infection and vaccination[35–39]. We did not, however, find a significant change in CD4$^{+}$CD25$^{+}$CD127$^{low}$ regulatory T cells

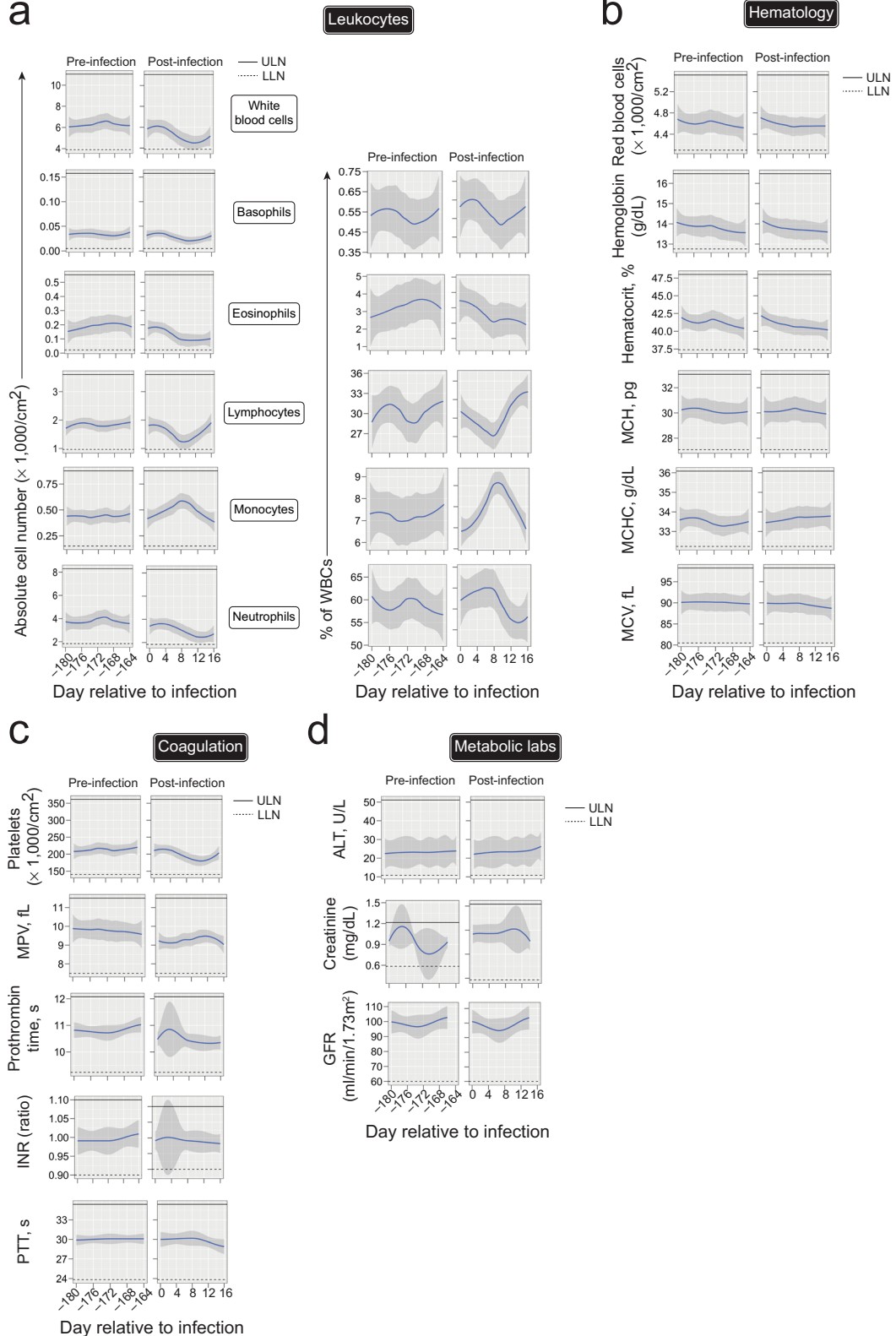

(Fig. 4g), which could be due to the deconvolution data using multiple genetic markers of Tregs and the possibility of missing bona-fide FOXP3+ cells due to our reliance on surface staining. In the B cell compartment, rDEN2Δ30 infection led to a small drop in naive B cells, which could also be reflected in a concurrent apparent rise in plasmablasts and switched memory B cells (Fig. 4h). Together our results indicate a bi-phasic immune activation with

early and sustained myeloid activation persisting after clearance, with the appearance of specific CD4+ and CD8+ memory T-cell populations following viremia.

**Logical conjunction approach to rDEN2Δ30-induced immunotranscriptome responses.** To reduce the complexity of the differential gene expression data set we applied logical conjunction

**Fig. 2 Clinical laboratory assessment of rDEN2Δ30 infection. a** Leukocyte populations expressed as absolute counts, (×1000 cells/cm$^2$, left) and as a percent of white blood cells (right) are shown for study visits occurring every other day over two ~2-week periods: 6 months preinfection and after infection of flavivirus-naive healthy volunteers ($n = 11$) with rDEN2Δ30, including day of infection. **b** Hematology labs, including red blood cell counts, serum hemoglobin, hematocrit, mean corpuscular hemoglobin (MCH), MCH concentration (MCHC), and mean corpuscular volume (MCV) are shown for pre- and post-infection as in (**a**). **c** Coagulation labs, platelet counts, mean platelet volume (MPV), Prothrombin time, international normalized ratio (INR), and partial thromboplastin time (PTT) are shown for pre- and post-infection as in (**a**). **d** Metabolic labs including alanine aminotransferase (ALT), Creatinine, and glomerular filtration rate (GFR) are shown for pre- and post-infection as in (**a**). Data shown for all subjects are shown as non-parametric LOESS (LOcal regrESSion) smoothing. All data also plotted by individual subject in Source Data. ULN upper limit of normal (solid lines), LLN lower limit of normal (dashed lines). For all panels, LOESS-smoothed lines and 95% confidence intervals are shown. Source data are provided as a Source Data file.

analysis (LCA) to identify a consensus transcript expression pattern shared by all individuals during rDEN2Δ30 infection. In LCA, for each timepoint comparison (e.g., days 8 vs. 0) we found all genes that significantly changed ($P < 0.05$ and FDR $< 0.1$) in a single subject and then asked which of these genes changed in the same directions for this time series in the next subject and so on, until a consensus set of gene expression changes for each comparison (days 8 vs. 0, days 28 vs. 8, and days 28 vs. 0) was reached and applied to all 11 subjects (Fig. 5a). This approach substantially reduced the number of differentially regulated genes compared to conventional timepoint-grouped analysis. Using LCA we found 151, 111, and 4 genes undergo regulation for days 8 vs. 0, 28 vs. 8 and 28 vs. 0 respectively (Fig. 5b). Greater than 99% of the genes that underwent regulation for LCA underwent regulation based on the grouped analysis. We found that 74 of the genes regulated between days 8 vs. 0 were the same genes that were regulated between days 28 and 8 (Fig. 5c). Since this pattern (up, then down) was congruent with the pattern of viremia during these timepoints we termed these Viremia-Tracking genes. The regulated genes from these comparisons that did not adhere to the viremia-tracking pattern (days 8 vs. 0, 60 upregulated and 16 downregulated; and Days 28 v 8, 37 downregulated), together with the 4 unique genes downregulated between days 28 vs. 0 comprised 118 genes termed Post-viremia genes. This term refers to the finding that these genes were changed during viral infection but did not return to baseline after resolution of viremia by day 28 (Fig. 5c). These data indicated the presence of a suite of genes directly regulated in parallel with viremia and a gene set exhibiting residual regulation and/or specifc association with the post -viremia state. Gene ontology pathway analysis revealed that the viremia-tracking set of genes was enriched for both response to and regulation of type I and II interferon pathways including JAK/STAT signaling. Also in this set were genes encoding for proteins that directly inhibit viral genome replication and those involved in protein ubiquitination and catabolism. In line with this, the most enriched pathway comprised genes in the interferon stimulated gene 15 (ISG15) pathway (Fig. 5d), a ubiquitin-like modifier involved in inhibition of viral replication[40]. The set of genes that were found to be altered 28 days after infection compared to baseline (i.e., post-viremia genes) encoded for more varied pathways including protein ubiquitination, cell migration, cytoskeletal reorganization, and angiogenesis. Concordant with the stimulation of antibodies by rDEN2Δ30 infection[41] humoral immune pathways were enriched after resolution of viremia. Innate immune gene expression (such as macrophage colony stimulating factor and type I/II interferon pathways) was changed 28 days after viral infection compared to baseline (Fig. 5e), which was unexpected given that WBCs including monocytes had returned to baseline within 2 weeks after infection). These results identified sets of genes whose regulation closely correlated with DENV2 viremia as well as those that did not return to baseline even after viremia was absent.

*Genes associated with rDEN2Δ30-induced immune activation.* Several clinical laboratory findings are associated with severe

natural dengue infection such as decreased WBC (mainly neutro- and lymphopenias) and indicators of potential coagulopathy including reduced platelets and altered partial thromboplastin time[2]. In the 1930's United States Army investigators detailed the transient, though sometimes profound, changes in the leukocyte compartment during experimental dengue infection[42]. In our primary rDEN2Δ30 infection model we noted transient decreases in neutrophil and lymphocyte counts increased monocytes (Fig. 3). We also noted a mild downward trend in PTT within normal limits and consistent with each patients' preinfection profile. We then leveraged this detailed clinical lab information against our longitudinal rDEN2Δ30-induced gene expression data to determine whether baseline gene expression would predict dengue-associated lab features.

Overall, we identified three groups containing a total of 49 genes that distinguished serological and cellular changes associated with rDEN2Δ30 infection (Fig. 6a). Group I genes ($n = 14$) involved regulation of type I interferons, adaptative immune-cell co-stimulation and differentiation, and ubiquitination (Fig. 6b) were positively associated with serum antibodies, and negatively associated with viremia and blood cell responses (Fig. 6a). Group II genes ($n = 19$) are largely involved in B cell and CD4 T-cell activation, differentiation, and function (Fig. 6c). This is concordant with our flow cytometric data and given the seroconversion to DENV2 after infection, with the role of genes in this group with induction of antibodies. Group II is primarily distinguished from Group I by positive correlation with lymphocyte and monocyte alterations compared with negative correlations for these features with Group I genes. Group II pathways were also associated with neutrophil nadirs and peak DENV2 viremia as were Group I genes (Fig. 6a). Group III genes ($n = 16$) (Fig. 6d) that primarily involved innate immunity, chemokine production and cellular metabolism were positively associated with changes in the leukocyte compartment. These changes were mirrored by negative associations with antibodies and peak viremia (Fig. 6a). Taken together our results reveal distinct sets of genes and pathways that demarcate rDEN2Δ30 viremia and neutralizing antibodies but can also be leveraged to understand homeostasis of the blood compartment during infection.

**Gene modules to predict rDEN2Δ30-associated rash.** The only clinical feature of rDEN2Δ30 infection was a rash consisted of a few maculo-papular lesions on the proximal upper extremities and chest that was generally unnoticed by the subjects and self-resolved in 5–10 days[29]. In the transcriptomics cohort, nine of 11 (82%) rDEN2Δ30-infected subjects exhibited a rash and two did not (Fig. 7a). Both rash and non-rash subjects were viremic for DENV2, but no specific features of viremia such as peak titer, duration, onset were associated with rash development. In fact, rash development occurred after day 8 in most rash subjects (6 of 9) so the RNA-seq data could not be matched to day of onset. We therefore asked whether it was possible to dichotomize rash versus non-rash based on baseline gene expression. To approach this we used linear

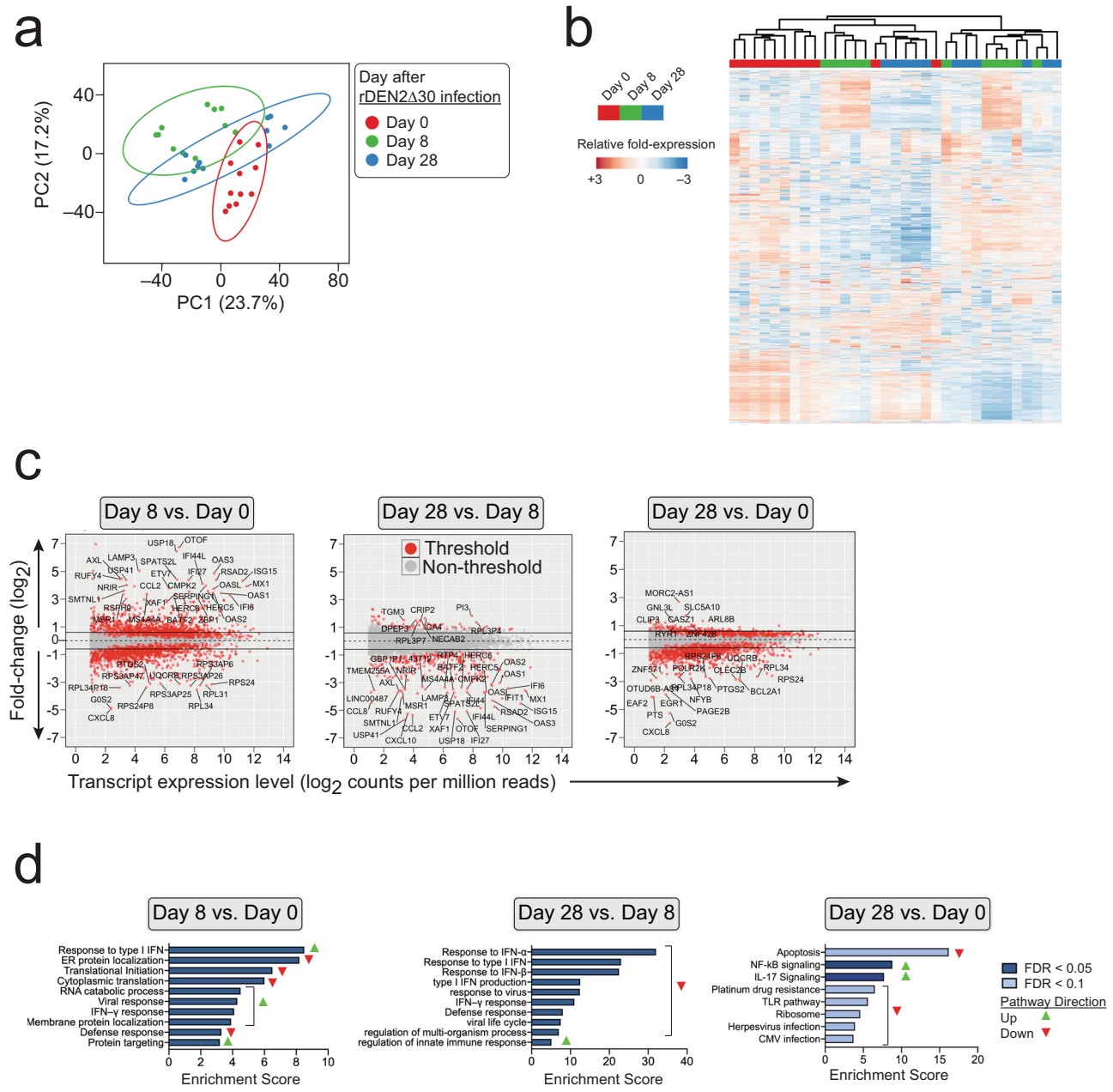

**Fig. 3 Group analysis of temporal gene expression analysis after rDEN2Δ30 primary infection. a** Principal component analysis by timepoint. One symbol per subject. **b** The top 2400 variable genes clustered on heatmap with column annotations for timepoints post infection. TMM values ln(x + 1)-transformed. Rows are centered; unit variance scaling is applied to rows. Both rows and columns are clustered using correlation distance and average linkage. **c** F × E plots (log₂ fold-change × expression, indicated by log₂ counts per million (CPM) reads) of pairwise timepoint comparisons. Differential gene expression threshold is significance level of $P < 0.05$, FDR < 0.1, and transcript level ≥ 4 CPM. Examples of highly regulated genes are labeled (**d**) Pathway analysis of differentially expressed genes in pairwise comparisons by timepoint after infection. Predicted pathway directionality (green up-pointing triangle, upregulated; red down-pointing triangle, downregulated) was determined by the behavior of the genes exhibiting |≥1.5-fold change|. Source data are provided as a Source Data file.

separability analysis[43] to identify 80 genes for which the baseline absolute transcript count was significantly different (i.e., differentially expressed) in the nine individuals that did experience rash versus those two that did not (Fig. 7a). We found 12 baseline whole blood-expressed genes that were linearly separable by at least 20 transcript counts that distinguished subjects that would go on to develop rash (or not) after rDEN2Δ30 infection (Fig. 7b). The top rash-identifying gene myeloid nuclear differentiation antigen (MNDA1) had a ΔCPM > 250, was more highly expressed in the whole blood of non-rash versus rash subjects. In line with this,

when assessing all rash-distinguishing genes, the pathways we found to be involved regulated myeloid responses, but also membrane regulation, autophagy, K63 ubiquitination, and cell morphogenesis (Fig. 7c). Our results suggest that some or all of these genes we identified may be useful as potential preinfection markers of risk of dengue-associated clinical features.

**Gene expression in severe dengue versus controlled rDEN2Δ30 infection.** Recently a 20-gene set derived from meta-analysis

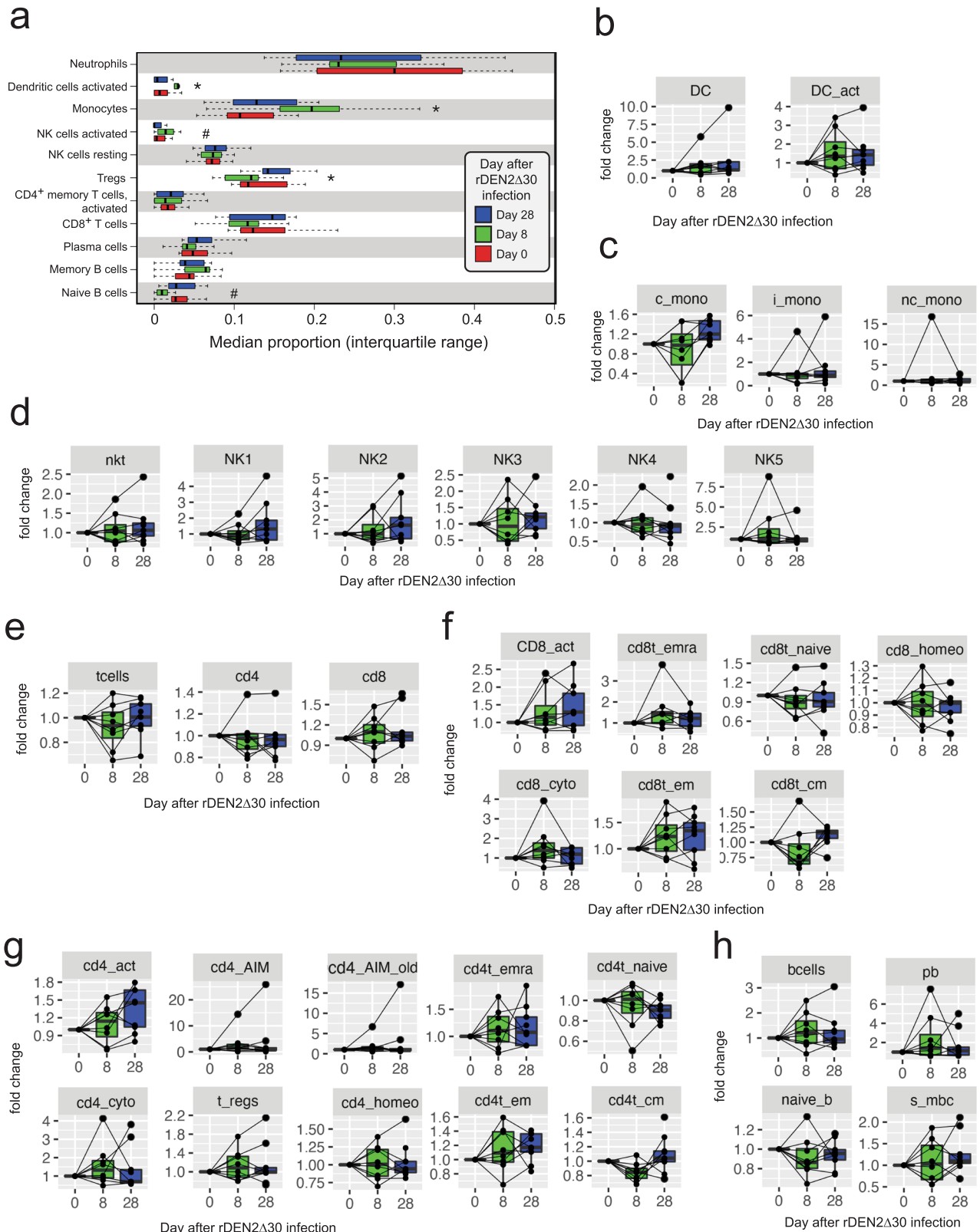

of genome-wide blood transcriptomic profiles from was found to predict progression to severe dengue in natural infection[20]. We next asked whether the genes regulated during a mild rDEN2Δ30 infection overlapped with those found to be regulated during severe dengue. Only one gene family, the guanine binding protein (GBP1/2) genes were regulated in severe dengue and during mild rDEN2Δ30 infection (Fig. 8). These results suggested that rDEN2Δ30 infection induced a largely distinct set of genes compared with those identified as regulated during severe dengue.

## Discussion

It is currently not possible to determine whether a DENV infection becomes symptomatic, the range of symptoms, and ultimately

**Fig. 4 Immune-cell dynamics before, during, and after rDEN2Δ30 infection. a** Deconvolution of gene expression using a leukocyte gene signatures matrix to obtain median proportions of select adaptive and innate white blood cells at each timepoint. Center lines show the medians; box limits indicate the 25th and 75th percentiles; whiskers represent 95% confidence interval. Kruskal–Wallis one-way ANOVA was used to determine $P$ values. $^*P < 0.05$; $^\#P < 0.1$. **b–g** Flow cytometric analysis of peripheral blood mononuclear cells from rDEN2Δ30-infected subjects. Data are expressed as fold change in cell frequency at days 8, and 28 after infection relative to day 0 baseline. Center lines show the medians; box limits indicate the 25th and 75th percentiles and all data points are shown. Kruskal–Wallis one-way ANOVA was used to determine $P$ values indicated as $^*P < 0.05$ and $^\#P < 0.1$. **b** Lineage-negative (CD3−CD19−CD56−CD14−CD16−) HLA-DR+ dendritic cells (DC), HLA-DR$^{hi}$ activated DC (DC_act); **c** monocytes: classical (c_mono), intermediate (i_mono), and non-classical (nc_mono); **d** natural killer T cells (nkt), NK1-5 populations (defined by CD56 and CD16 staining as outlined in Supplementary Fig. 4); **e** CD3+ T cells (t cells), CD4+ T cells (cd4), and CD8+ T cells (cd8); **f** CD8+ T-cell populations: activated CD279+ (cd8_act), cytotoxic CD57+ (cd8_cyto), CCR7−CD45RA+ T-effector re-expressing CD45RA (cd8t_emra), CCR7+CD45RA+ naïve (cd8t_naive), CCR7−CD45RA− effector memory (cd8t_em), CCR7+CD45RA− central memory (cd8t_cm), and homeostatic CD127+ (cd8_homeo); **g** CD4+ T-cell populations: activated CD279+ (cd4_act), cytotoxic CD57+ (cd4_cyto), activation-induced marker (AIM)-positive CD154+CD134+(cd4_AIM), CD25+CD134+ (cd4_AIM_old), regulatory T cells CD127−CD25+ (t_regs), homeostatic CD127+ (cd4_homeo), CCR7−CD45RA+ T effector re-expressing CD45RA (cd4t_emra), CCR7+CD45RA+ naïve (cd4t_naive), CCR7−CD45RA− effector memory (cd4t_em), CCR7+CD45RA− central memory (cd4t_cm); **h** CD19+ B cells (bcells), plasmablasts CD19+ CD38$^{hi}$CD27$^{hi}$ (pb), IgM+CD27− naïve B cells (naive_b), and IgM−CD27+ switched memory B cells (s_mbc). See Fig. S4 for gating. Source data are provided as a Source Data file.

progression to severe dengue. Biomarkers for each of these stages would be useful for public health and vaccination development strategies. For dengue whole blood transcriptomics have been employed to identify potential biomarkers for risk of severe disease[10–18,44], but not for asymptomatic[1,3] infection. To address this, we leveraged whole blood RNA samples collected during experimental infection of subjects with a partially attenuated recombinant rDEN2Δ30 challenge virus. We recovered live virus from the serum in 100% of infected subjects, though at levels far below those observed in ongoing severe dengue disease[45]. rDEN2Δ30 infection elicited DENV2-neutralizing antibodies, most of which are DENV2 serotype-specific[41]. Temporally, we show here that viremia onset begins within 4 days after infection and neutralizing antibodies are present 1 month after infection and persist for months afterward. Infection with rDEN2Δ30 led a transient non-pruritic rash in 80% of subjects[29].

To explore gene expression over time after rDEN2Δ30 infection we first used a canonical approach to group all subjects by timepoint and determine median differential gene expression as a function of time by generalized linear model. rDEN2Δ30 infection triggered dynamic transcriptional regulation in the systemic immune compartment. The grouped analysis was also useful in the deconvolution of the whole blood transcriptional response to rDEN2Δ30 infection revealed transient changes in activated DC and monocytes, NK cells, and naive B cells. Natural infection studies identifying myeloid cells as key targets of DENV and Zika virus (ZIKV) replication[46–48] and recently NK cells and B cells have been also associated with viral transcription[19]. We confirmed several of these targets including myeloid-derived monocytes and activated DCs by flow cytometry and this analysis revealed other increased in adaptive immunity in the T-cell compartment including memory populations known to be regulated during dengue infection or vaccination[35–39].

rDEN2Δ30 infection led to early induction of antiviral control mechanisms such as reduction in ribosomal subunit genes (*RPL*, *RPS*) and activation of eukaryotic translation initiation factor 2 subunit α (eIF2α)-regulated genes to suppress translation and viral assembly[49]. Some interferon stimulated genes such as *OAS1*, *OAS3*, *MX1, and RSAD2* that were strongly induced by rDEN2Δ30 infection are also known to be regulated in natural infection[14,16]. Other ISGs regulated by rDEN2Δ30 infection such as *ISG15* and *IFI27* have not been well-documented in natural infection though may play a role in controlling viral replication among other functions[40].

To generate a concise and potentially generalizable signature of rDEN2Δ30 infection we developed a LCA approach to identify DEGs that changed in the same direction at the same timepoints

after infection in all subjects. We validated that all the LCA DEGs were present in the grouped analysis and defined a group of 74 genes that tracked with viremia (i.e., went up during viremia and then back to baseline). ISG15 protein conjugation was the most prominent pathway regulated in this fashion. This protein modification may be involved in control of DENV infection[50] and is involved in response to other viruses[40]. In addition, 117 genes did not follow the pattern of viremia—76 of these were regulated by infection, but did not return to baseline after viral clearance; 41 post-viremia genes were specifically regulated only during virus clearance. These post-viremia DEGs functionally clustered in pathways humoral immunity, immune regulation (such as control of IFN-β signaling), cellular metabolism, regulation of platelets, and potential for cellular migration. Some examples of such genes are *C1QA/B* (complement pathway), *CD300C* (NK cell regulation), *IL27* (T-cell differentiation), *LAG3* (immune regulation), *IFI44/L, IFIT1/2, IFIH1, and IRF7* (all interferon-inducible genes). While several of these (such as the interferon-inducible pathways) also known to be regulated during DENV infection, a role in post-viremic responses has not yet been described.

Our individual subject-level LCA facilitated predictive gene expression signatures associated with several systemic characteristics of rDEN2Δ30 infection. We correlated baseline gene expression with peak viremia, neutralizing antibodies, neutrophil and platelet nadirs as well as myeloid and lymphoid cell behaviors after rDEN2Δ30 infection. Likewise, Chan et al. found unique preinfection baseline transcriptional profiles corresponding to perturbations in endoplasmic reticulum stress and cellular metabolism in subjects who later experienced clinical symptoms (mainly fever) after receiving the yellow fever 17D vaccine (YF17DV)[51]. Kwissa et al. and Popper et al. have found key roles for myeloid and type interferon response with development of neutralizing antibodies after natural infection and vaccination with rDENV3Δ30/31, respectively[12,52].

A rash, usually lasting more than a week, is a common, but not absolute, clinical feature of dengue, but has unknown prognostic capacity. The only clinically important feature of rDEN2Δ30 infection is a transient, non-pruritic macropapular rash in 80% of subjects[29]. No features of viremia (titer, duration, etc.) were associated with development of rash, so to gain insight into baseline predictors for potential for rash after DENV infection we assessed baseline gene expression as a function of subsequent rash development. As in the YF17DV study[51] our work makes use of a true preinfection baseline which allowed linkage of baseline gene expression prediction to clinical outcomes. For rDEN2Δ30 infection, higher baseline expression of myeloid nuclear differentiation antigen (*MNDA*), and cell surface associated cellular processes such

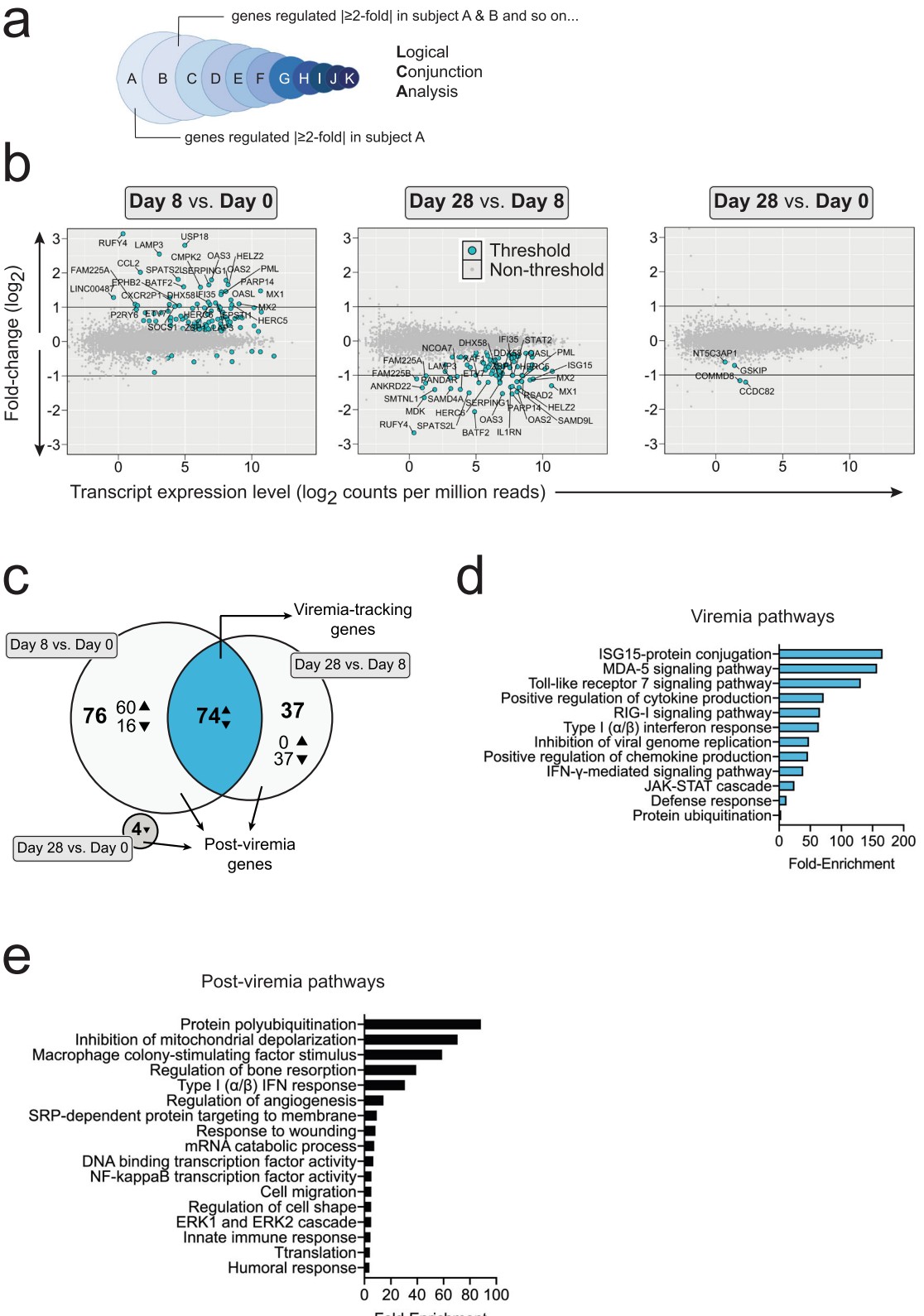

**Fig. 5 Temporal gene expression analysis after rDEN2Δ30 infection by logical conjunction analysis (LCA). a** Schematic overview of LCA. Genes have undergone regulation if for all 11 subjects the regulation occurred in the same direction (e.g., up) and had a *P* value < 0.05 and FDR < 0.1. No minimum fold-change criteria was applied for Threshold. **b** F × E plots (log2 fold-change × expression, in counts per million reads (CPM)) of pairwise timepoint comparisons for genes meeting threshold of *P* < 0.05 and FDR < 0.1 and ≥4 counts per million reads (CPM). Examples of genes regulated |≥1.5-fold| are labeled (**c**) Venn diagram of gene expression changes by timepoint. Triangles indicate directionality of gene regulation in timepoint comparison. **d** DAVID Pathway analysis of Viremia-Tracking genes (**e**) DAVID pathway analysis of post-viremia genes. Directionality is not shown because the pathway definition incorporates three timepoints. Source data are provided as a Source Data file.

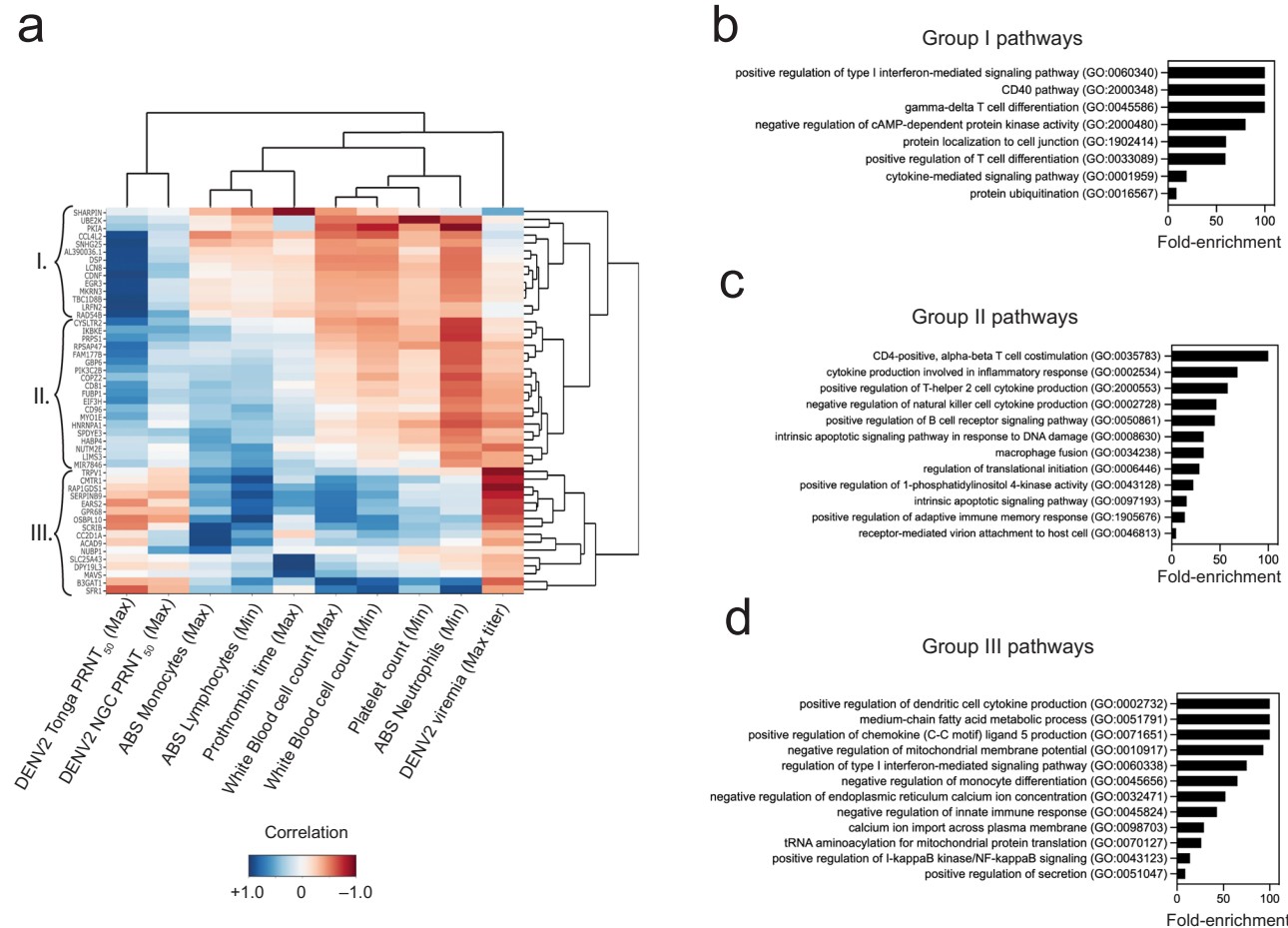

**Fig. 6 Association of preinfection baseline gene expression with clinical features of rDEN2Δ30 infection. a** Genome-wide absolute baseline gene expression level counts (rows) were queried against ten clinical laboratory parameters (columns) associated with rDEN2Δ30 infection including peak viremia, neutralizing antibodies and changes in blood cell counts across the 11 subjects. Only genes (rows) that had an absolute correlation >0.9 for at least one of the columns are pictured. Darker colors represent a stronger correlation; while blue and red represent positive and negative correlation, respectively. Unsupervised clustering identified 49 genes and three major groups (I–III) distinguishing amongst the parameters. **b–d** Pathway analysis and GO terms associated with Group 1 (**b**), Group II (**c**), and Group III (**d**) gene associations and unadjusted P values with FDR < 0.1 are shown. Source data are provided as a Source Data file.

as tetraspanin *CD37*, integral membrane 2B (*ITM2B*), and genes involved in autophagy (*VMP1*) was associated with protection from rash. Potential mechanisms for this could involve altered myeloid migratory capacity to the skin, which has been shown to occur in mice in response to DENV infection[47]. Our results indicate a subject-specific conjunctive approach has predictive value for associating blood gene expression with systemic outcomes related to rDEN2Δ30 infection.

Lastly, we compared all DEGs we identified by LCA with a recently published 20-gene set to predict severe dengue[20] to investigate whether severe dengue DEGs may be an amplification of those found in mild DENV infection such as rDEN2Δ30, or a distinct profile. The 20-gene set was stringently selected from 8 whole blood genome-wide transcriptomics data sets of severe dengue. Likewise, our LCA was stringent in that it narrowed the minimum gene sets changing in all subjects after rDEN2Δ30 infection. One of the viremia-tracking genes we identified – guanylate binding protein (GBP)1—is in the same gene family as *GBP2* from the severe dengue gene set. GBPs are interferon-induced genes located on the 1p22.1 region of Chromosome 1 and the protein products with direct antiviral activity[53] and may be a common component of dengue infection and pathogenesis. Nonetheless, the overall difference in these gene sets suggested

distinct transcriptome events regulated by mild rDEN2Δ30 infection compared to hospitalized patients progressing to severe dengue. An important caveat is that our study does not capture the natural variability in the kinetics of the progression to severe dengue so we cannot formally exclude the possibility that the genes regulated by rDEN2Δ30 could be regulated at similar timepoints during severe dengue and further work may clarify this point.

There were several other limitations to our study. We studied a relatively small cohort of 11 subjects, only two of whom were truly asymptomatic and did not have a rash. To address our sample size we used two approaches to DEG analysis, including a FDR < 0.1 cutoff and the LCA approach required that DEG patterns were directionally similar and correspond to the phenotype analyzed in every subject at the same timepoint. Another limitation is the use of whole blood which does not identify specific cell type transcriptional changes that occur, as has been done in individual patients by single cell RNA Seq[19]. We were limited in our analysis of baseline DEG analysis to predict rash by having only 2 subjects without rash. Within the context of our experimental rDEN2Δ30 infection model, LCA was able to identify rash-predicting gene expression. Future work may determine gene associations, if any, with wild type dengue rash. Our DEG

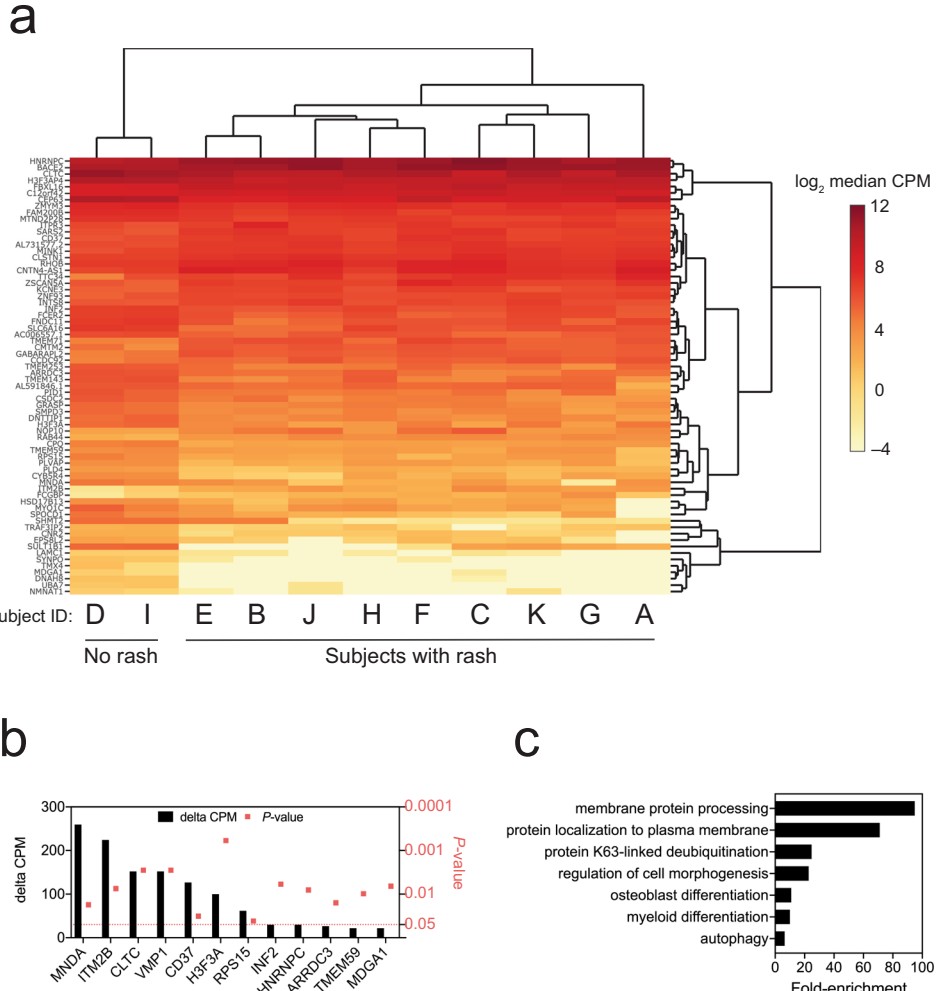

**Fig. 7 Association of preinfection gene expression with rash following rDEN2Δ30 infection. a** Hierarchical clustering of transcript levels (expressed) counts per million (cpm) for genes that have a day 0 cpm that is linearly separable for subjects that have rash and those that do not have rash. Darker shades of red indicate a higher median expression ($log_2 cpm$) on day 0. **b** Genes that distinguish rash versus non-rash based on linear separability greater than 20 (i.e., the median Δ cpm for rash versus non-rash is |≥20|) with corresponding $P$ values (square symbols). **c** DAVID pathway analysis for all significant rash-distinguishing linear separable genes ($n = 56$). Source data are provided as a Source Data file.

analysis timepoints did not overlap exactly with each event measured (rash, viremia, antibodies, WBC changes, etc.) which changed over a longer time interval, finer temporal resolution may be needed to temporally correlate gene expression and systemic rDEN2Δ30-related immune responses. Overall, our results define potentially generalizable gene sets modulated during a clinically mild rDEN2Δ30 infection which correlate with systemic immune changes and appear to be distinct from those occurring during severe dengue disease.

## Methods

**Human subjects**. Volunteers were infected with $10^3$ focus-forming units of rDEN2Δ30 (ClinicalTrials.gov registration no. NCT02021968) as part of the control (i.e., placebo) arm of a vaccine/challenge study to evaluate the efficacy of a tetravalent dengue vaccine against challenge with rDEN2Δ30[29]. Volunteers were recruited from the Baltimore, MD and Burlington, VT at Johns Hopkins University Center for Immunization Research and the University of Vermont Vaccine Testing Center, respectively under Institutional Review Board approval granted at both institutions. Informed consent included use of blood samples for research with the text "*genetic differences in responses to vaccines or dengue virus infection (samples will be used anonymously)*". Subject IDs are coded here as A-K.

**Clinical procedures, evaluation of adverse events**. Study procedures and criteria for adverse events such as rash have been described[28,29,54,55]. Clinical Laboratory Improvement Amendments (CLIA)-accepted upper and lower limits of normal

(ULN, LLN) for blood lab values were used. Non-parametric LOESS (LOcal regrESSion) was used for smoothing in this data shown in Fig. 2. At study visits, blood was collected by venipuncture into serum separator tubes for analyses of viremia and serology, and into EDTA tubes for isolation of peripheral blood mononuclear cells (PBMC). Serum was frozen at −20 °C until use. PBMC were isolated by Ficoll-paque density gradient separation, counted, and frozen in cell culture medium with 10% dimethyl sulfoxide (DMSO) and 40% fetal bovine serum (FBS), and cryopreserved in liquid nitrogen vapor phase.

**Measurements of viremia and neutralizing antibodies**. Viremia was evaluated at days 0, 4, 6, 8, 10, 12, 14, and 16 after rDEN2Δ30 infection as described[55,56]. Antibody testing was evaluated in serum samples at days 0, 28, 56, 90, and 180. Neutralizing antibody titers to DENV2 Tonga74 (American genotype) and against the New Guinea C (NGC) strain (Asian genotype II) (Supplementary Table 2) were measured by plaque reduction (by 50%) neutralization titers (FRNT$_{50}$) on Vero81 cells[55,56].

**RNA isolation and RNA-seq procedures**. Whole blood (2.5 mL) was collected from test subjects using PAXgene Blood RNA tubes containing 6.9 mL of additive (PreAnalytix/Qiagen cat. No 762165) on days 0, 8, and 28 post rDEN2Δ30. For some day 0 samples (for subjects A-E), 0.8 mL of whole blood was aliquoted into 2.3 mL of PAXgene additive and this preparation was split into 2 aliquots of 1.55 mL each. PAXgene-preserved blood was stored at −80 °C. Samples were thawed and total RNA was extracted, and DNase treated using the PAXgene Blood kit equipped with RNA spin column 1017507 (PreAnalytiX cat. no. #763134, Hombrechtikon, Switzerland) from 9.4 mL or ~2 mL (for day 0 samples for subjects A, B, C, and E). Resulting RNA was protected by adding Ribolock RNase inhibitor (ThermoFisher, Waltham, MA USA, cat. no. EO0381) to a final concentration of

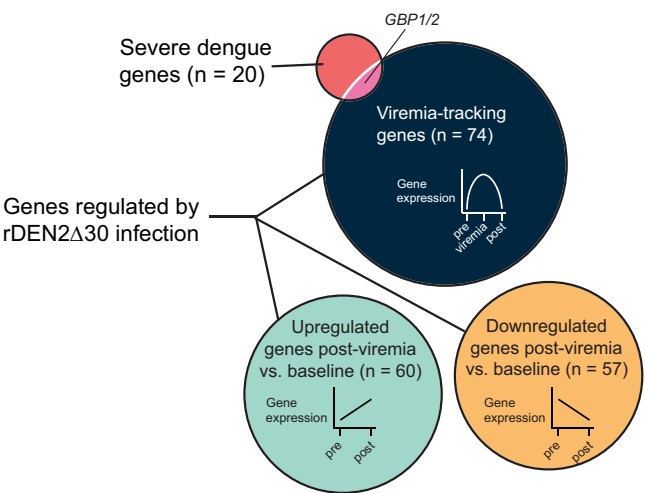

**Fig. 8 Distinct gene regulation in different phases of rDEN2Δ30 infection and in severe dengue.** Venn diagram of genes associated with severe dengue from Robinson et al. (ref. [20]) in relation to viremia-tracking genes (n = 74) and to those that were upregulated (n = 60) or downregulated (n = 57) post-viremia compared to baseline following rDEN2Δ30 infection. Source data are provided as a Source Data file.

0.25 U/μL of RNA. RNA was evaluated for quality based on 28 S and 18 S rRNA using the RNA Nanochip (Agilent cat. no. 5067–1512) on the Agilent Bioanalyzer 2100 (Santa Clara, Ca). The 260/280 ratio was assessed on a Nanodrop ND1000 (ThermoFisher Waltham, MA) and quantified using the RNA HS kit (Thermo-Fisher cat. no. Q32852) on the Qubit spectrofluorometer (ThermoFisher Waltham, MA). RNA-seq libraries for Illumina sequencing were constructed using the NuGEN Ovation Universal RNA reagent with ribosomal RNA and globin depletion (cat. no. 0338-32, NuGEN-Tecan, Redwood City, CA).

To generate technical replicates as a key factor in the bootstrapping analysis method we used to analyze gene expression data we loaded equal amounts of each sample onto three different lanes of the flow-cell. There was <1% variation in total number of reads among the lanes for any sample. Sequencing of $1 \times 100$ bp single-end reads was performed on a HiSeq 2000 using V3 reagents (Illumina). The Illumina sequences were trimmed of bases with a Phred quality score <15 and any contaminating adapters used in cDNA and sequencing library preparation. Only single-end reads which survived trimming and were ≥60 bases in length were mapped to the human (GRCh37, version 17, Ensembl 72) genome using stringtie[57]. Raw reads were normalized to library DNA input and then mapped to the Human reference Genome 38 using stringtie. Mean mapped library sizes ($\times 10^6$ reads) were $26 \pm 0.6$ standard deviations [min–max: 14.4–35], $28 \pm 0.7$ [17.8–40], and $31 \pm 0.9$ [15–40] across the day 0, day 8, and day 28 timepoints respectively. The minimum and maximum library sizes for any subject/timepoint combination was $14$–$46 \times 10^6$ mapped reads with a mean of $28.5 \times 10^6$. This range meets the 2017 ENCODE guidelines for transcriptome sequencing depth ($10$–$25 \times 10^6$ mapped reads per library at 100 bp single-end sequencing-by-synthesis)[58].

### RNA-seq analysis

*Grouped analysis.* We first assessed DEG during rDEN2Δ30 infection by grouping all subjects by timepoint to capture a maximal number of potential genes and pathways. Raw counts were imported and analyzed using the edgeR package[59,60] and gene names were extracted using ballgown[61] in R (version 4.0.2). Counts were adjusted for library size and normalized using the using the TMM (trimmed mean of M values) method. Counts were retained only if a gene counts per million reads (CPM) were ≤2 and occurring in at least three samples. Dispersion and differential expression were determined using the QL F-test GLM option in edgeR, which was chosen to account for patient heterogeneity, non-normal distribution of cells within blood samples, and limited sample size for this study. The design model of $\sim 0 + \text{day}$ was used and individual contrasts (day 0 vs. 8, day 8 vs. 28, and day 0 vs. 28) were made using the glmQLFTest() and considered significant at FDR < 0.01.

Principal component analysis and heatmap in Fig. 2a, b were generated with Clustvis using the top 2400 most variable genes[62]. TMM normalized values were ln (x + 1)-transformed and unit variance scaling was applied to rows. Singular value decomposition with imputation was used to calculate the principal components. Prediction ellipses are such that with probability 0.95, a new observation from the same group will fall inside the ellipse.

*Inference of immune-cell type abundance and pathway analysis.* BAM files were merged by sample using SAMtools[63] and counts were generated at the gene level using featureCounts[64]. Immune-cell proportions were analyzed from merged BAM

files by CIBERSORT against LM22 gene signatures[32] using the full expression matrix (Transcript Per Million (TPM) >10). Tissue-specific signatures such as mast cells and macrophages were censored for application to blood.

*Logical conjunction analysis.* We leveraged the three sequencing lanes for every subject at each timepoint to apply LCA to the RNA-seq data set. We further assumed that blood drawn from any given subject is a representative (i.e., homogeneous) sample for that subject[65]. Our rationale is that any gene that undergoes gene regulation across all eleven subjects at a given time interval is more likely to be associated with the response to rDEN2Δ30 infection when compared to measuring average gene regulation for all eleven subjects grouped by timepoint. To generate intra-sample/subject gene expression measurement error critical to this approach we assessed gene levels separately in each of the three sequencing lanes run per sample/timepoint. The mRNA count data set was preprocessed such that only genes with at least two counts per million (for at least three replicates) were analyzed for gene expression. We used edgeR[59,60] to determine differential gene expression between days 8 and 0, 28 and 8, and 28 and 0. Genes were considered to be differentially expressed if all genes increased or decreased in the same direction for all 11 subjects at a given time interval; and for each subject, the differential expression for a gene had a $p$ value < 0.05 and a false discovery rate < 0.1[66]. To determine the number of genes expected to change in the same direction at a given time interval, Markov chain Monte Carlo was run using a million permutations where gene regulation from the three different timepoints were randomly shuffled across genes and subjects. Assuming no dependence between the infection and gene regulation, the Markov chain Monte Carlo 99% confidence intervals for the number of genes expected to undergo gene regulation is [0, 1]. Therefore, there is a high degree of confidence that the regulated genes are related to rDEN2Δ30 infection. We used KEGG and DAVID to assign biological functions to the sets of genes that were differentially regulated across different timepoints. Pathway enrichment analysis was done primarily with DAVID[67] and confirmed with Webgestault[68] and PantherDB[69].

### Flow cytometry

Cryopreserved PBMC were thawed in Iscove's Modified Dulbecco's medium supplemented with 8% (v/v) FBS in PBS and rested overnight at 37 °C. On day of sample staining, cells were washed in PBS + 1% FBS and counted. For each sample, $1 \times 10^6$ cells were resuspended in 50 μL of 1% FBS in (PBS) and stained with Live/Dead Blue Viability Dye (0.5 μL/test, Invitrogen, catalogue #L23105A) and treated with Fc and monocyte blocker (Human Trustain FcX blocker, 5 μL/test, Biolegend, catalogue #422302) according to the manufacturers' instructions. The antibody cocktail was prepared in 50 μL of Brilliant stain buffer (BD Horizon) and comprised the following: anti-human CD3 (UCHT1, FITC-conjugated, 0.25 μL/test, BioLegend, catalog #300406), anti-human CD4 (OKT4, BV510-conjugated, 1 μL/test, Biolegend, catalogue #317444), anti-human CD8 (RPA-T8, BV650-conjugated, 0.5 μL/test, Biolegend, catalogue #301041), anti-human CD14 (M5E2, BV711-conjugated, 1 μL/test, Biolegend, catalogue #301837), anti-human CD16 (3G8, APC-Cy7-conjugated, 0.25 μL/test, Biolegend, catalogue #302017), anti-human CD19 (HIB19, PE-Dazzle594-conjugated, 0.5 μL/test, Biolegend, catalogue #302252), anti-human CD25 (M-A251, BV421-conjugated, 0.5 μL/test, Biolegend, catalogue #356113), anti-human CD27 (M-T271, PE-Cy7-conjugated, 0.125 μL/test, Biolegend, catalogue #356412), anti-human CD38 (HIT2, Alexa Fluor 647-conjugated, 0.125 μL/test, Biolegend, catalogue #303514), anti-human CD45RA (HI100, BUV395-conjugated, 0.25 μL/test, BD OptiBuild, catalogue #740298), anti-human CD56 (NCAM16.2, BUV563-conjugated, 0.125 μL/test, BD Horizon, catalogue #612929), anti-human CD57 (QA17A04, BV605-conjugated, 0.5 μL/test, Biolegend, catalogue #393303), anti-human CD127 (HIL-7R-M21, BUV805-conjugated, 0.25 μL/test, BD OptiBuild, catalogue #748486), anti-human CD134 (ACT35, BUB737-conjugated, 0.125 μL/test, BD OptiBuild, catalogue #749286), anti-human CD154 (24-31, BV785-conjugated, 0.5 μL/test, Biolegend, catalogue #310841), anti-human HLA-DR (L243, BV570-conjugated, 2.5 μL/test, Biolegend, catalogue #307637), anti-human IgM (MHM-88, PerCP-Cy5.5-conjugated, 0.5 μL/test, Biolegend, catalogue #314512), anti-human CCR7 (2-L1-A, APC-R700-conjugated, 0.5 μL/test, BD Horizon, catalogue #566767), and anti-human CD279 (EH12.2H7, PE-conjugated, 0.5 μL/test, Biolegend, catalogue #329905). Samples were stained in the antibody cocktail in the dark for 30 min at 4 °C, then washed twice with FACS buffer (1% FBS in PBS). All samples were acquired on a Cytek Aurora. The staining panel (was designed in collaboration with Dr. Roxana del Rio-Guerra at the UVM flow cytometry facility using a pan-leukocyte Optimized Multicolor Immunofluorescence Panel (OMIP)- 024 (ref. [70]). Neutrophils were not analyzed due to their paucity in Ficoll-prepared PBMCs. In addition, we included markers to identify recently activated T cells with CD279 (PD-1) or those harboring activation-induced marker (AIM) by two different strategies (CD25+CD134+ [OX40], AIM_old)[71] or CD134+CD154 [CD40L], AIM)[72]. The antibodies and reagents used in this study are listed in Supplementary Table 2. The gating strategy is Supplementary Fig. 4 and the titration of each staining antibody is Supplementary Fig. 5.

**Reporting summary.** Further information on research design is available in the Nature Research Reporting Summary linked to this article.

### Data availability

RNA Seq data is on the National Center for Biotechnology Information (NCBI) Gene Expression Omnibus under accession number GSE152255. All other data are available in

the Article file, Supplementary Information or available from the authors upon reasonable request. Source data are provided with this paper.

## Code availability

All computations and quantifications were performed using the R programming language. Custom scripts can be found at: https://github.com/seandiehl/uvm-vaccine-lab/tree/seandiehl-ncomms_paper or https://doi.org/10.5281/zenodo.4552689.

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

## Acknowledgements
We thank the study volunteers and clinical staff at the University of Vermont (UVM) Vaccine Testing Center and the Johns Hopkins Center for Immunization Research. We thank Pheobe Laaguiby and Jess Hoffman of the University of Vermont Integrative Genomics Resource (VIGR) for technical expertise in preparation of samples. We thank Roxana del Rio-Guerra for assistance in design and optimization of the flow cytometry panel. We also thank Joe Boyd and Adelaide Rhodes for assistance in analysis. This work was funded by a pilot grant from the UVM Larner College of Medicine and College of Engineering and Mathematical Science (to S.A.D., S.V.S., and D.M.R.), by the National Institute of Health (P20GM125498 to B.D.K., Project 4 to S.A.D. and U01AI141997 to S.A.D. and B.D.K.) and the Bill and Melinda Gates Foundation (OPP1109415). The next-generation sequencing was performed in the VIGR Massively Parallel Sequencing Facility and was supported by the UVM Cancer Center, Lake Champlain Cancer Research Organization, UVM College of Agriculture and Life Sciences, and the Larner College of Medicine, and in part by National Institutes of Health (NIH) grant P30-GM118228. The University of Vermont Bassett Flow Cytometry and Cell Sorting Facility and supported by NIH grants S10-ODO18175 and P30GM118228 with the Cytek Aurora *supported by S10-ODO026843.* The clinical trial was supported by the National Institute of Allergy and Infectious Diseases (NIAID) Intramural Research Program contract HHSN272200900010C.

## Author contributions
S.A.D. and S.V.S. conceived the study with additional design input from J.H. and D.M.R. S.T. performed RNA-seq experiments. J.H., J.A.D., K.M.E., S.F., and S.A.D. performed bioinformatics analysis, H.A.T. performed flow cytometric experiments and analysis. R. DR.-G. designed and optimized the flow cytometry panel. D.M.D. and N.S. collected and organized clinical data, S.S.W., A.P.D., K.K.P., B.D.K. provided specimens and oversaw the clinical trial. S.A.D. and D.M.R. supervised the project. S.A.D. wrote the paper with input from co-authors. All authors have approved the submitted version.

## Competing interests
The authors declare no competing interests.
