## [Peer Review File · Nature Communications]

REVIEWER COMMENTS

Reviewer #1 (Remarks to the Author):

This manuscript by Hanley and colleagues report the transcriptomic changes that occur in humans after a primary experimental infection with a partially attenuated dengue type 2 virus (DENV-2). The motivation of this study was to define the host response that shaped primary dengue that, according to the authors, mostly result in asymptomatic infection. The authors leveraged on a clinical trial that evaluated the efficacy of a dengue vaccine candidate using a controlled human infection model; these volunteers were all seronegative for dengue at baseline. The authors thus explored in the unvaccinated population how primary infection with the DENV-2 challenge strain affected the host response to infection longitudinally. They identified sets of genes that tracked with viremia and those which expression were sustained even after viremia levels waned. The authors also found genes expressed at pre-infection baseline that correlated with various infection outcome, including white cell count, lymphocyte count and the appearance of rash. Finally, the authors compared their findings with those reported from prior transcriptomic studies in severe dengue patients and found mostly no overlap in the genes that were differentially expressed. They suggested that their findings represent generalizable gene sets that affect asymptomatic DENV-2 infection.

The question on what host response differentiates asymptomatic from symptomatic infection is important. It is a major gap in knowledge in the field of dengue. The topic of investigation is thus highly interesting. Unfortunately, there are several limitations to this study, chiefly because of how this study was designed. These are:

1. The most concerning limitation is that only 2 of the 11 subjects included in this study had true symptomatic infection. The remaining 9 out of 11 reported a rash following infection and were thus not truly asymptomatic. The identification of the host transcriptional response that differentiates asymptomatic from symptomatic infection necessitates the comparison between true asymptomatic subjects with those that develop symptoms after infection with the same virus at the same inoculum (as was done for yellow fever vaccine – Chan et al, Nat Med 2019;25:1218-1224). Framing this study as an attempt to define the host response in asymptomatic DENV-2 infection is thus highly problematic – the authors had to rely on the notion that most primary DENV-2 infection is asymptomatic and, by extension, argue that their findings are representative of asymptomatic infection. Instead, my suggestion is for the authors to consider framing this study as an approach to define the longitudinal host response to DENV infection. That the authors were able to show the transcriptional level changes prior to symptom onset is already a major contribution to the knowledge on dengue pathogenesis. It would in no way, at least to this reviewer, diminish the significance of the contribution of this study to the field. As it is and with such a small number of truly asymptomatic subjects, the data that differentiates symptomatic from asymptomatic infection is very preliminary, making the discussion and conclusion of the paper speculative.
2. Line 59. The statement on primary DENV infection being mostly asymptomatic ignores differences in infection outcome from the 4 DENV serotypes. Please revise this statement to capture the DENV type differences in infection outcome (Clapham et al, PLoS Negl Trop Dis 2015;9:e0004262).
3. Lines 60-62. This statement suggests that the host response that determines disease severity is universal for all four types of DENV infection. This may not be the case as indicated in the Clapham et al reference given above, which was also expressed by Vaughn in Am J Epidemiol 2000;152:800-803. This statement should be revised to reflect this nuance.
4. Lines 91-119. Much of the narrative here should be in the results and not the methods section.
5. Line 97 and Supplemental File 1. One of the subjects appear to have had a badly hemolyzed blood sample. Would this present problem for the transcriptomic analysis?
6. Lines 103-105 and Supplemental File 1 – white cell count. The WCC at day 0 appears to be clustered into 2 populations, one group at ~4K/cmm and another at ~8K/cmm. Can the authors comment on this

clustering? This is especially curious since the patients were said to be selected for analysis based on their viremia profile.

7. Lines 280-282. The authors indicated that innate immune genes were residually changed after infection. This is somewhat of an overreaching statement as the study only monitored the blood transcriptome for 28 days post infection. Whether the baseline was reset by infection or whether complete resolution of innate immune activation was longer than expected has not been clearly defined.

8. Lines 332-340. A major problem with comparing transcriptome from this study against those that in patients with wild-type DENV infection is the time from infection. Since the host response to infection is dynamic, controlling for the time from infection is critical to avoid misleading conclusion. This analysis has not limited the analysis to the equivalent of day 8 from infection in patients. The concluding statement based on the data in Fig 7 alone is speculative.

Reviewer #2 (Remarks to the Author):

The authors of this study provide a transcriptomic analysis of peripheral blood from patients experimentally infected with an attenuated strain of DENV2.

While there have been many studies of the consensus transcriptional profile of peripheral blood in dengue patients, one advantage of the study is that the subjects were experimentally infected. This allows data from early time points and precisely matched time points, which are often unavailable from studies of patients with naturally acquired infection. However, considering this is a greatly attenuated strain that is used and not a wild-type virus, there are caveats whether this response truly represents an asymptomatic dengue infection and this needs to be acknowledged in the introduction/discussion. Where it is written "dengue infection" it should be changed to "DENV2 Δ 30 infection"

Due to the fact that transcriptomics data was done as consensus on whole blood they need to use an algorithm to deconvolute the likely cell responses. However, the limitations of this algorithm are apparent since they identify mast cells in the blood when there are no mast cells in circulation. This emphasizes that it would be important to validate some of the suggested populations by flow cytometry.

They identified genes and pathways that persist into the convalescent period following infection. The case for the post-infection signature being "myeloid regulatory" is not strongly made since other pathways associated with the IFN response, humoral response and tissue repair are also quite strongly regulated. The identification of rash-associated genes is more interesting, particularly since it is the "pre-infection" transcriptional response that is identified, but was it not identified whether any acute phases responses could also distinguish rash and no-rash groups. Do they? That said, there is a concern whether having only 2 patients in the no-rash group is sufficient for analysis.

The manuscript would benefit greatly from a better description of the clinical course of infection in this study cohort and inclusion of the associated data for laboratory measures in the main figures rather than supplemental.

Line 290- High viremia has not always been associated with severe dengue; there are conflicting reports.

Figure 1 legend needs to say that they are experimentally infected. What are the neutralizing titers to the inoculating strain? Some error in wording line 608. How is Fig. 1 different from previously published data on this cohort?

Figure 7, this presentation is unusual and the colors aren't very clear. Could this be represented alternatively?

It is not clear what is meant by "imprinted" and how this determination was made.

Discussion line ~420, there are other reasons why this data set could differ from severe patients in other data sets including technical differences and analyses, besides just the aspect of asymptomatic disease. Again, this was meant to be a vaccine candidate initially and is not the same as true naturally acquired infection with strains that can be either mild or severe in different patients. Also, since 80% of subjects experienced rash, it is not well justified that the disease represented was truly asymptomatic.

Reviewer #3 (Remarks to the Author):

The manuscript by Hanley et al analyze the blood gene expression patterns induced by infection of humans with the attenuated rDENV230 virus. Infection caused a largely asymptomatic infection, characterized by mild rash in some subjects. The results reveal that inflammatory genes including type I interferon and viral restriction pathways were induced during DENV2 viremia and returned to baseline after viral clearance, while others including myeloid, migratory, humoral, and growth factor immune regulation factors pathways were found at non-baseline levels post-viremia. Furthermore, the authors analyzed pre-infection baseline gene expression and report that a baseline signature could predict DENV2-induced immune responses and development of rash. The results are interpreted to suggest that distinct immunological profiles for severe and asymptomatic dengue and offer new potential biomarkers for characterizing primary DENV infection.

The subject of this paper is of general interest. Whilst several previous studies have analyzed transcriptional profiles in natural dengue infection, the current study is one of the first to study transcriptional response to a controlled human challenge model with dengue. The advantage of this is that, unlike previous studies that have relied on samples from convalescent patients as a "baseline," in the current study the authors use a bona fide baseline by analyzing PBMCs from day 0.

However, the paper in its current form, suffers from several weaknesses including over interpretation of the data, which cast doubt on the strength of the conclusions that can be drawn:

1. The extent to which the results obtained from this controlled human challenge model with an attenuated strain of DENV2, can be extrapolated to a natural infection is unclear. The title of the paper "A myeloid regulation signature persists after virus clearance in mild human primary dengue serotype 2 infection," is misleading because it conveys the impression that this study was performed in natural dengue infections. Indeed, the viral loads shown in Supplemental Fig 1 are much lower than those that have been described in natural infection. So the title should be revised to reflect this major caveat, as follows: "A myeloid regulation signature persists after virus clearance in a controlled human primary infection challenge model with dengue serotype 2."

In addition, the abstract and discussion should be modified accordingly.

2. The cohort of 11 subjects analyzed in this study were chosen from 20 subjects in the study. By what criteria were these subjects chosen? The relatively small sample size raises doubts about the statistical robustness of the findings.

3. The aforementioned cohort comprises 2 females and 9 males, and roughly 50% black and white subjects. Can the authors provide some analysis to see if there was any association of signatures with sex or race?

4. The authors use the word "asymptomatic" to describe the clinical symptoms in this study, but the mechanisms underlying the "asymptomatic" response in the subjects in the current study with the attenuated DENV2, could be very different from those that mediate asymptomatic infection during natural infection. This caveat should be acknowledged in the abstract, introduction and discussion.

Response to Reviewers

We thank all referees for their time and thoughtful input on our study. We have now revised the manuscript and have responded to all queries therein and in this response. Line numbers for changes in the manuscript made in response to referees' comments are indicated in each response below. This manuscript has been substantially improved by the referees' suggestions.

Reviewer #1 (Remarks to the Author):

This manuscript by Hanley and colleagues report the transcriptomic changes that occur in humans after a primary experimental infection with a partially attenuated dengue type 2 virus (DENV-2). The motivation of this study was to define the host response that shaped primary dengue that, according to the authors, mostly result in asymptomatic infection. The authors leveraged on a clinical trial that evaluated the efficacy of a dengue vaccine candidate using a controlled human infection model; these volunteers were all seronegative for dengue at baseline. The authors thus explored in the unvaccinated population how primary infection with the DENV-2 challenge strain affected the host response to infection longitudinally. They identified sets of genes that tracked with viremia and those which expression were sustained even after viremia levels waned. The authors also found genes expressed at pre-infection baseline that correlated with various infection outcome, including white cell count, lymphocyte count and the appearance of rash. Finally, the authors compared their findings with those reported from prior transcriptomic studies in severe dengue patients and found mostly no overlap in the genes that were differentially expressed. They suggested that their findings represent generalizable gene sets that affect asymptomatic DENV-2 infection.

The question on what host response differentiates asymptomatic from symptomatic infection is important. It is a major gap in knowledge in the field of dengue. The topic of investigation is thus highly interesting. Unfortunately, there are several limitations to this study, chiefly because of how this study was designed. These are:

1. The most concerning limitation is that only 2 of the 11 subjects included in this study had true symptomatic infection. The remaining 9 out of 11 reported a rash following infection and were thus not truly asymptomatic. The identification of the host transcriptional response that differentiates asymptomatic from symptomatic infection necessitates the comparison between true asymptomatic subjects with those that develop symptoms after infection with the same virus at the same inoculum (as was done for yellow fever vaccine – Chan et al, Nat Med 2019;25:1218-1224). Framing this study as an attempt to define the host response in asymptomatic DENV-2 infection is thus highly problematic – the authors had to rely on the notion that most primary DENV-2 infection is asymptomatic and, by extension, argue that their findings are representative of asymptomatic infection. Instead, my suggestion is for the authors to consider framing this study as an approach to define the longitudinal host response to DENV infection. That the authors were able to show the transcriptional level changes prior to symptom onset is already a major contribution to the knowledge on dengue pathogenesis. It would in no way, at least to this reviewer, diminish the significance of the contribution of this study to the field. As it is and with such a small number of truly asymptomatic subjects, the data that differentiates symptomatic from asymptomatic infection is very preliminary, making the discussion and conclusion of the paper speculative.

We appreciate the reviewer's overall support of the study. We agree that our longitudinal study including a baseline, reproducible viremia, and differential dengue mild symptomology can contribute to our understanding of dengue pathogenesis. As suggested by the reviewer, we

have reframed the study as the response to an attenuated challenge rDEN2Δ30 virus. We have modified the discussion and conclusions accordingly so as not to represent our study as a model of an asymptomatic DENV infection. We agree that Chan et al. offers a useful comparison of a live attenuated flavivirus vaccine to our live attenuated non-vaccine candidate and has now been cited (ref. 51) and discussed in Lines 347-350 and 359-361.

2. Line 59. The statement on primary DENV infection being mostly asymptomatic ignores differences in infection outcome from the 4 DENV serotypes. Please revise this statement to capture the DENV type differences in infection outcome (Clapham et al, PLoS Negl Trop Dis 2015;9:e0004262).

The reviewer is correct and we thank them for reminding us of this reference. It has now been cited as ref. 21 and a statement has been added (Line 63-64).

3. Lines 60-62. This statement suggests that the host response that determines disease severity is universal for all four types of DENV infection. This may not be the case as indicated in the Clapham et al reference given above, which was also expressed by Vaughn in Am J Epidemiol 2000;152:800-803. This statement should be revised to reflect this nuance.

We have captured this in our modified statement on serotype-specific infections and have rephrased this part of our rationale (Lines 63-66) away from asymptomatic vs. symptomatic infection in the spirit of the aforementioned use of the 'rDEN2Δ30 infection' terminology. We also focus on the role of the virus rather than the host in our Introduction (lines 97-99).

4. Lines 91-119. Much of the narrative here should be in the results and not the methods section.

We have moved this text to the Results and made a new Figure 2 summarizing the clinical labs. We have retained the subject-specific findings in the Supplement.

5. Line 97 and Supplemental File 1. One of the subjects appear to have had a badly hemolyzed blood sample. Would this present problem for the transcriptomic analysis?

We are not clear on which subject and by which metric the reviewer is referring. This is speculating, but Subject C's corpuscular RBC metrics were always below normal range, but their total hemoglobin (Hb) was always in normal range (~13-16.5 g/dL) whereas severe hemolysis is typically defined as Hb >30 g/dL. None of the subjects appears to present as having in vivo hemolysis since all Hbs were all normal. I've had a pathologist from our hospital's clinical lab review the Supplemental data and they concur that hemolysis cannot be directly concluded from this data.

In our clinical workflow there are separate tubes for the RNA and lab analyses. If a lab tube hemolyzed because of handling, we can't say if that also affected the RNA tube (which contains lysis buffer itself). RNA tubes are frozen immediately after collection and initial processing. Finally, we deplete all globin transcripts (hemoglobin, myoglobin, plakoglobin, etc) from RNA sample prior to transcriptional analysis (see lines 439 and 139). We thank the reviewer for this opportunity to review our clinical data and processes.

6. Lines 103-105 and Supplemental File 1 – white cell count. The WCC at day 0 appears to be clustered into 2 populations, one group at ~4K/cmm and another at ~8K/cmm. Can the authors

comment on this clustering? This is especially curious since the patients were said to be selected for analysis based on their viremia profile.

Indeed, we had not noted that clustering and have now gone back and looked at viremia as a function thereof– the peak level of those in the “8K/cmm cluster was 2.0 log₁₀ PFU/mL [1.75-2.6 range] and those in the 4K/cmm cluster was 2.2 [2 – 2.8 range] PFU/mL. The P-value for this comparison was 0.41. We will note this in the results (page 7, line 114-118).

7. Lines 280-282. The authors indicated that innate immune genes were residually changed after infection. This is somewhat of an overreaching statement as the study only monitored the blood transcriptome for 28 days post infection. Whether the baseline was reset by infection or whether complete resolution of innate immune activation was longer than expected has not been clearly defined.

We have clarified in lines 219-220 that the definition of “after infection” was after detectable viremia was gone (by Day 28 in all subjects). The reviewer raises an excellent point that innate, perhaps myeloid, resolution is not yet complete by Day 28. We have modified this passage (lines 232-235).

8. Lines 332-340. A major problem with comparing transcriptome from this study against those that in patients with wild-type DENV infection is the time from infection. Since the host response to infection is dynamic, controlling for the time from infection is critical to avoid misleading conclusion. This analysis has not limited the analysis to the equivalent of day 8 from infection in patients. The concluding statement based on the data in Fig 7 alone is speculative.

We agree with this point and have revised our results statement to: “These results suggested that controlled rDEN2Δ30 infection induced a largely distinct set of genes compared with those identified as regulated during severe dengue.” (Lines 294-295)

Furthermore, in the Discussion we also address the reviewer’s caveat about wild-type infection kinetics and the limitations of our model: “An important caveat is that our study does not capture the natural variability in the kinetics of the progression to severe dengue so we cannot formally exclude the possibility that the genes regulated by rDEN2Δ30 could be regulated at similar timepoints during severe dengue and further work may clarify this point.” (lines 379-382).

Reviewer #2 (Remarks to the Author):

The authors of this study provide a transcriptomic analysis of peripheral blood from patients experimentally infected with an attenuated strain of DENV2.

1. While there have been many studies of the consensus transcriptional profile of peripheral blood in dengue patients, one advantage of the study is that the subjects were experimentally infected. This allows data from early time points and precisely matched time points, which are often unavailable from studies of patients with naturally acquired infection. However, considering this is a greatly attenuated strain that is used and not a wild-type virus, there are caveats whether this response truly represents an asymptomatic dengue infection and this needs to be acknowledged in the introduction/discussion. Where it is written “dengue infection” it should be changed to “DENV2Δ30 infection”

We have made this change and refer only to “rDEN2Δ30” throughout.

2. Due to the fact that transcriptomics data was done as consensus on whole blood they need to use an algorithm to deconvolute the likely cell responses. However, the limitations of this algorithm are apparent since they identify mast cells in the blood when there are no mast cells in circulation. This emphasizes that it would be important to validate some of the suggested populations by flow cytometry.

We thank the reviewer for pointing this out. This indeed a limitation of deconvolution analysis. Our chosen approach (CIBERSORT) was developed for analysis of tissues (which can contain mast cells or macrophages as opposed to monocytes), which may explain our initial result. We have now clarified our rationale for this approach and that our re-analysis censored tissue-resident cells like mast cells. We also now note that CIBERSORT was used for used to investigate immune gene expression during malaria infection (now cited as ref. 33 and discussed in lines 167-168, 476-477). Our analysis of cells in the blood by WBC with differential (New Figure 2) indicated that monocytes (i.e. myeloid cells) are strongly regulated by rDEN2Δ30 infection. This was validated by our CIBERSORT analysis, giving us confidence in this approach. We also compared two other deconvolution methods, both of which failed to validate our data showing regulation of monocytes during rDEN2Δ30 infection. To further address the reviewer's inquiry we performed flow cytometry on paired samples from 9 of the 11 donors for which we had PBMCs at days 0, 8, and 28 after infection. In a new Figure 4 and in the Results (lines 179-200), we confirmed that monocytes (particularly classical CD16⁺CD14⁻ IL-10–producing monocytes as well as activated DCs are elevated after viral clearance following rDEN2Δ30 infection. Our flow panel also captured changes in the adaptive immune compartment, mainly in activation of CD4 and CD8 T cell memory. These findings are also discussed in Lines 317-320.

3. They identified genes and pathways that persist into the convalescent period following infection. The case for the post-infection signature being “myeloid regulatory” is not strongly made since other pathways associated with the IFN response, humoral response and tissue repair are also quite strongly regulated. The identification of rash-associated genes is more interesting, particularly since it is the “pre-infection” transcriptional response that is identified, but was it not identified whether any acute phases responses could also distinguish rash and no-rash groups. Do they? That said, there is a concern whether having only 2 patients in the no-rash group is sufficient for analysis.

We appreciate this opportunity to reflect on this in light of our new flow cytometry data. Our deconvolution analyses identified myeloid cells including monocytes and activated DCs as being modified by dengue infection and we confirmed this by flow cytometric analysis, particularly the elevated post-viral classical monocytes. However, we could not recapitulate the Tregs changes identified by deconvolution analysis in our flow cytometry data. This is explainable by limited marker availability in flow compared to multi-transcript identification of bioinformatic analysis. Although pathway analysis independently revealed both myeloid and regulatory features of the convalescent transcriptome, we feel the Treg flow data diminished our ability to make claims about prolonged immune regulatory effects, especially given the turnover of or differentiation of monocytes out of the blood. We feel our new title captures the novelty of our study and better represents our findings. Again, we thank the reviewer for prompting us to examine these cell populations by flow cytometry.

Regarding the rash question, we focused on pre-infection transcriptome as a predictor of our principal clinical finding. We had explored whether the single acute phase timepoint could also be related to the rash, but our Day 8 timepoint for RNA available did not line up with kinetics of

rash onset (See Table R1) and this precluded us from concluding of the relationship of acute phase gene expression signatures with onset or development of rash. This is clarified in lines 272-276).

Reviewers Table 1 Rash onset	Days 0 - 8	Days 9 - 28
Number of rDEN2Δ30 subjects with rash onset in the following timeframes after infection	3	6

We agree that only having two patients in the no-rash group limits a more general statement on whether patterns associated with gene expression for the rash and no-rash groups would persist given more patients. However, each patient has three technical replicates which is sufficient to demonstrate, that given the threshold $P < 0.05$, there is a difference between the distribution of each no-rash subject with each rash subject for the genes we highlighted. Our rationale in highlighting these genes is to provide the reader with a list of genes that could be investigated for association with rash in natural dengue patients (lines 390-392).

The manuscript would benefit greatly from a better description of the clinical course of infection in this study cohort and inclusion of the associated data for laboratory measures in the main figures rather than supplemental.

We have moved this text to the Results and have presented a new Figure 2, which presents summarized clinical data. Individual subject data are still in Supplemental File 1.

Line 290- High viremia has not always been associated with severe dengue; there are conflicting reports.

We have removed this statement.

Figure 1 legend needs to say that they are experimentally infected. We have modified the Figure legend to denote experimental infection with rDEN2Δ30 Tonga virus. What are the neutralizing titers to the inoculating strain? These are shown in Fig. 1b. The addition of details on experimental infection (with strain info) in the legend clarify this. Some error in wording line 608. We did not find a line 608 and did not find clear errors in lines 108, 208, 308, or 508. How is Fig. 1 different from previously published data on this cohort? Figure 1 is different because detailed kinetics or subject-level viremia or antibody information were never published on this cohort. We note this in Results, lines 109-110.

Figure 7, this presentation is unusual and the colors aren't very clear. Could this be represented alternatively?

We have created a new clearer figure (Fig. 8) with improved color scheme and legend.

It is not clear what is meant by "imprinted" and how this determination was made.

We have abandoned this terminology and use "[Up- or down-] regulated genes post-viremia vs. baseline" in the legend. "Imprinted" is also removed from the results.

Discussion line ~420, there are other reasons why this data set could differ from severe patients in other data sets including technical differences and analyses, besides just the aspect of

asymptomatic disease. Again, this was meant to be a vaccine candidate initially and is not the same as true naturally acquired infection with strains that can be either mild or severe in different patients. Also, since 80% of subjects experienced rash, it is not well justified that the disease represented was truly asymptomatic.

This point was also raised by reviewer #1; we take this point and have reframed the study as a longitudinal analysis of gene expression during a mild experimental dengue infection and have made many changes in the text to reflect this.

Reviewer #3 (Remarks to the Author):

The manuscript by Hanley et al analyze the blood gene expression patterns induced by infection of humans with the attenuated rDENV230 virus. Infection caused a largely asymptomatic infection, characterized by mild rash in some subjects. The results reveal that inflammatory genes including type I interferon and viral restriction pathways were induced during DENV2 viremia and returned to baseline after viral clearance, while others including myeloid, migratory, humoral, and growth factor immune regulation factors pathways were found at non-baseline levels post-viremia. Furthermore, the authors analyzed pre-infection baseline gene expression and report that a baseline signature could predict DENV2-induced immune responses and development of rash. The results are interpreted to suggest that distinct immunological profiles for severe and asymptomatic dengue and offer new potential biomarkers for characterizing primary DENV infection.

The subject of this paper is of general interest. Whilst several previous studies have analyzed transcriptional profiles in natural dengue infection, the current study is one of the first to study transcriptional response to a controlled human challenge model with dengue. The advantage of this is that, unlike previous studies that have relied on samples from convalescent patients as a "baseline," in the current study the authors use a bona fide baseline by analyzing PBMCs from day 0.

However, the paper in its current form, suffers from several weaknesses including over interpretation of the data, which cast doubt on the strength of the conclusions that can be drawn:

1. The extent to which the results obtained from this controlled human challenge model with an attenuated strain of DENV2, can be extrapolated to a natural infection is unclear. The title of the paper "A myeloid regulation signature persists after virus clearance in mild human primary dengue serotype 2 infection," is misleading because it conveys the impression that this study was performed in natural dengue infections. Indeed, the viral loads shown in Supplemental Fig 1 are much lower than those that have been described in natural infection. So the title should be revised to reflect this major caveat, as follows: "'A myeloid regulation signature persists after virus clearance in a controlled human primary infection challenge model with dengue serotype 2.'"

We appreciate this criticism and completely agree. We agree and have changed the title of the manuscript to clearly reflect experimental rDEN2Δ30 infection.

In addition, the abstract and discussion should be modified accordingly. Also done.

2. The cohort of 11 subjects analyzed in this study were chosen from 20 subjects in the study.

By what criteria were these subjects chosen? The relatively small sample size raises doubts about the statistical robustness of the findings.

We have updated the Results to show that we selected based on having representative demographics and viremia compared to the full cohort (lines 93-99).

The LCA approach can provide robustness with smaller sample sizes since it leverages the differential expression of each individual subject. In the grouped analysis, we show that thousands of genes are differentially expressed for each of the three time periods. While using LCA, there is over a 10-fold decrease in the number of genes associated with dengue. The root cause of this difference is that the grouped analysis is not as robust. By aggregating the results of the subjects using the grouped analysis then genes are considered differentially expressed even if one of the subjects does not undergo differential expression or in some cases, may exhibit differential gene expression in the opposite direction of the other subjects. The reason for this is that the signal of an outlier subject (or subjects) is averaged out by the signal from other subjects. An example of this is demonstrated in the figure below.

In the figure above, the grouped analysis says that ARID1A underwent up regulation between days 0 and 8. However, the individual analysis says that subjects H, I, and J underwent differential expression in the downward direction during this time period and the other 8 subjects underwent differential expression in the upward direction. This is just one example where the signals from the majority of the subjects can result in a less than robust determination of differential expression. Adding more subjects to our analysis could possibly lessen the number of extraneous genes called using the grouped analysis, however, there will still be instances like ARID1A where a gene is determined have differentially expressed in one direction when in fact a minority of the subjects did not differentially express or differentially expressed in the opposite direction of the majority. Therefore, we are confident in the robustness of our results using LCA even though the number of subjects is small.

3. The aforementioned cohort comprises 2 females and 9 males, and roughly 50% black and white subjects. Can the authors provide some analysis to see if there was any association of signatures with sex or race?

We thank the reviewer for this suggestion because although we lose power by splitting our cohort into smaller groups we did notice some interesting trends that we are providing for the reviewer.

When we looked at development of rash, there were no significant patterns with respect to race and/or gender and development of rash. The two subjects that did not have rash were male but

given there were only two female subjects one cannot say that only males cannot get rash. Also, the two subjects that did not have rash are of different races, so there is no evidence that rash is related to race.

For viremia, there was no difference in peak viremia between White and Black subjects (Reviewer Figure 1). The mean day of peak viremia was earlier after infection for White subjects (6.3 ± 0.8 days) compared to that for Black subjects (9.2 ± 4.6 days). There was also more variability in day of peak viremia for the Black subjects compared to White subjects. This was assessed by calculating the Shannon entropy metric (where lower values correspond to less variability), with values of 0.65 vs. 1.92 for White and Black subjects, respectively. Of note, this was the same metric used to equalize the subset cohort variability in viremia to that of the parent cohort).

With respect to gender the mean and standard deviation for day of peak viremia is 7.6 ± 3.6 and 8.0 ± 2.8 for male and female subjects, respectively and this is not significant. The Shannon entropy it is 1.88 and 1.00 for the Male and Female subjects, respectively which is not a significant finding.

To investigate putative signatures for race or gender we performed LCA for whether the number of differentially expressed genes (DEG) for a comparison (race or biological sex) for distinct time intervals exceeded the 99% confidence interval (CI) of the maximum number that could change by chance for each comparison.

Reviewer Table 2. Number of DEG found by LCA in comparisons by race or gender by timepoint. Highlighted rows indicate comparisons for which the number of DEG returned which exceeded that which was expected by chance (with 99% confidence interval).

RaceOrGender	TimePeriod	Regulation	NumberOfGenes	99% CI
Black	Day 0 - 8	Up	22	[0, 8]
Black	Day 0 - 8	Down	19	[0, 9]
Black	Day 8 - 28	Up	1	[0, 7]
Black	Day 8 - 28	Down	1	[0, 8]
Black	Day 0 - 28	Up	24	[0, 8]
Black	Day 0 - 28	Down	15	[0, 10]
White	Day 0 - 8	Up	1	[0, 5]

White	Day 0 - 8	Down	4	[0, 5]
White	Day 8 - 28	Up	9	[0, 4]
White	Day 8 - 28	Down	20	[0, 4]
White	Day 0 - 28	Up	1	[0, 5]
White	Day 0 - 28	Down	5	[0, 5]
Male	Day 0 - 8	Up	1	[0, 1]
Male	Day 0 - 8	Down	1	[0, 1]
Male	Day 8 - 28	Up	0	[0, 1]
Male	Day 8 - 28	Down	0	[0, 1]
Male	Day 0 - 28	Up	0	[0, 1]
Male	Day 0 - 28	Down	4	[0, 1]
Female	Day 0 - 8	Up	48	[64, 112]
Female	Day 0 - 8	Down	45	[61, 108]
Female	Day 8 - 28	Up	74	[71, 124]
Female	Day 8 - 28	Down	33	[67, 117]
Female	Day 0 - 28	Up	72	[64, 113]
Female	Day 0 - 28	Down	67	[57, 103]

For race, we found that between days 0 and 8, there were 41 DEG where the regulation was overrepresented for Black versus White subjects. Black subjects upregulated interferon signaling, cell migratory, and the unfolded protein response to a greater extent and had reduced mitochondrial-associated apoptosis. Between days 0 and 28, Black subjects only exhibited increased pathways associated with protein phosphorylation and endothelial cell proliferation, but few genes were involved. Between Days 8 and 28 (i.e. during resolution of viremia) White subjects showed higher transcription factor activity in the B cell compartment (PAX5) and myeloid compartment (MZF1, myeloid zinc finger 1).

As for gender, only four genes were down-regulated between days 0 and 28 more than in female subjects, but this did not lend itself to pathway analysis.

Though we appreciate the opportunity to query race and gender we feel that inclusion of this analysis is beyond the scope of this study. Our goal was to identify immunotranscriptomes elicited by experimental rDEN2 Δ 30 infection with a focus on clinically relevant findings, none of which differed by race or gender. However, a deeper dive into the transcriptional differences in response to rDEN2 Δ 30 infection by race will be a topic of future studies.

4. The authors use the word "asymptomatic" to describe the clinical symptoms in this study, but the mechanisms underlying the "asymptomatic" response in the subjects in the current study with the attenuated DENV2, could be very different from those that mediate asymptomatic infection during natural infection. This caveat should be acknowledged in the abstract, introduction and discussion.

This point was also raised by other reviewers. We take this point and have reframed the study as a longitudinal analysis of gene expression during a mild experimental dengue infection and exclusively refer now to rDEN2 Δ 30 infection when discussing our results from this model.

REVIEWER COMMENTS

Reviewer #1 (Remarks to the Author):

I would like to thank the authors for the extensive revision they have done to their manuscript, in response to the reviewers' comments. The manuscript reads well and now adds a vital piece to the understanding of the host response to dengue virus infection. Congratulations on this outstanding study!

Reviewer #2 (Remarks to the Author):

The authors have addressed all of my comments and I think the manuscript has been improved.

Reviewer #3 (Remarks to the Author):

The authors have satisfactorily addressed my comments. I look forward to seeing this in print.

REVIEWERS' COMMENTS

Reviewer #1 (Remarks to the Author):

I would like to thank the authors for the extensive revision they have done to their manuscript, in response to the reviewers' comments. The manuscript reads well and now adds a vital piece to the understanding of the host response to dengue virus infection. Congratulations on this outstanding study!

Reviewer #2 (Remarks to the Author):

The authors have addressed all of my comments and I think the manuscript has been improved.

Reviewer #3 (Remarks to the Author):

The authors have satisfactorily addressed my comments. I look forward to seeing this in print.

- We thank all of the reviewers for their thorough review of this manuscript. We agree that it is much improved.